# MMOT: The First Challenging Benchmark for Drone-based Multispectral Multi-Object Tracking

**Tianhao Li**[1,2]    **Tingfa Xu**[1,2*]    **Ying Wang**[1]    **Haolin Qin**[1]    **Xu Lin**[1]    **Jianan Li**[1,2*]

[1]Beijing Institute of Technology
[2]Beijing Institute of Technology Chongqing Innovation Center
`ciom_xtf1@bit.edu.cn,lijianan15@gmail.com`
*Corresponding author

## Abstract

Drone-based multi-object tracking is essential yet highly challenging due to small targets, severe occlusions, and cluttered backgrounds. Existing RGB-based multi-object tracking algorithms heavily depend on spatial appearance cues such as color and texture, which often degrade in aerial views, compromising tracking reliability. Multispectral imagery, capturing pixel-level spectral reflectance, provides crucial spectral cues that significantly enhance object discriminability under degraded spatial conditions. However, the lack of dedicated multispectral UAV datasets has hindered progress in this domain. To bridge this gap, we introduce **MMOT**, the first challenging benchmark for drone-based multispectral multi-object tracking dataset. It features three key characteristics: (i) **Large Scale** — 125 video sequences with over 488.8K annotations across eight object categories; (ii) **Comprehensive Challenges** — covering diverse real-world challenges such as extreme small targets, high-density scenarios, severe occlusions, and complex platform motion; and (iii) **Precise Oriented Annotations** — enabling accurate localization and reduced object ambiguity under aerial perspectives. To better extract spectral features and leverage oriented annotations, we further present a multispectral and orientation-aware MOT scheme adapting existing MOT methods, featuring: (i) a lightweight Spectral 3D-Stem integrating spectral features while preserving compatibility with RGB pretraining; (ii) an orientation-aware Kalman filter for precise state estimation; and (iii) an end-to-end orientation-adaptive transformer architecture. Extensive experiments across representative trackers consistently show that multispectral input markedly improves tracking performance over RGB baselines, particularly for small and densely packed objects. We believe our work will benefit the community in advancing drone-based multispectral multi-object tracking research. Our MMOT, code and benchmarks are publicly available at `https://github.com/Annzstbl/MMOT`.

## 1 Introduction

Unmanned aerial vehicles (UAVs) serve as a versatile platform for multi-object tracking (MOT) in dynamic, large-scale environments, supporting applications in surveillance [1], search and rescue [2], and aerial delivery [3]. In practice, drone-based MOT faces several significant challenges, including the low resolution of distant objects, high density of targets, and complex background. Conventional RGB-based tracking algorithms primarily rely on spatial appearance features for object detection and association such as shape, color, and texture. Yet in such challenging aerial scenarios, these features become severely degraded or indistinct, leading to reduced discriminability for object tracking, as shown in Fig. 1(a), where pedestrians are visually indistinguishable from the background. Therefore,

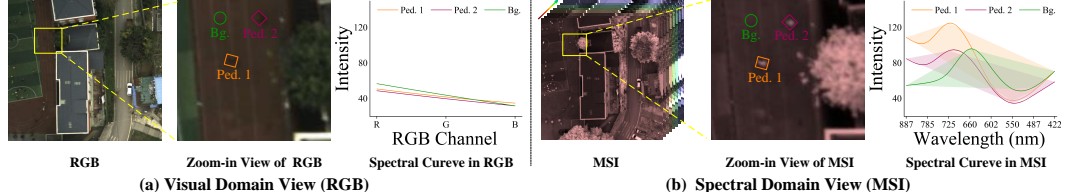

| RGB | Zoom-in View of RGB | Spectral Cureve in RGB | MSI | Zoom-in View of MSI | Spectral Cureve in MSI |
|---|---|---|---|---|---|
| **(a) Visual Domain View (RGB)** | | | **(b) Spectral Domain View (MSI)** | | |

Figure 1: Targets are more distinguishable in multispectral imagery. In the RGB view (left), pedestrians are visually indistinct and overwhelmed by background. In contrast, the MSI view (right) reveals clear spectral separation between targets and the background, highlighting the enhanced discriminability provided by spectral cues.

it is imperative to explore complementary feature dimensions beyond spatial appearance to enhance target separability and improve both the accuracy and robustness of drone-based multi-object tracking.

Multispectral imaging (MSI) captures both spatial and spectral cues, enabling per-pixel spectral measurements that reveal object properties beyond visual appearance and provide a more informative scene representation than RGB. The spectral dimension offers complementary cues that improve object discrimination and association, especially under small objects and cluttered backgrounds. As shown in Fig. 1, pedestrians are visually indistinct from the background in RGB due to small size and similar color. In contrast, MSI reveals clear spectral differences, as confirmed by the distinct spectral curves, enabling improved target-background separability. Therefore, compared to conventional RGB imagery, multispectral data provide a more effective solution for object tracking by introducing a complementary and discriminative spectral dimension. However, the absence of dedicated datasets for drone-based multispectral multi-object tracking presents a significant gap, limiting the development and evaluation of advanced methods in this emerging domain.

To bridge the gap, this work presents MMOT, the first large-scale and challenging multispectral UAV MOT dataset. The dataset is collected using a drone-mounted multispectral camera with a downward-facing view, capturing real-world urban scenes across varying dates, flight altitudes, and weather conditions. The dataset feature three key characteristics:

- **Large Data Scale.** The dataset comprises 125 video sequences totaling 13.8K frames, captured at a spatial resolution of $1200 \times 900$ with 8 spectral bands spanning from visible to near-infrared range. It includes 488.8K annotated bounding boxes, which are manually labeled, requiring over 5,000 work hours, thus ensuring high-quality annotations and providing a solid foundation.

- **Comprehensive Challenging Attributes.** During data collection, challenges encountered by drone-based MOT in real-world scenarios were carefully considered including extremely small targets, densely packed instances, severe occlusions, fast object motion and irregular UAV motion. Such conditions naturally arise in practical applications and collectively reflect the complex conditions that robust tracking systems must contend with.

- **Precise Oriented Bounding Box Annotation.** Due to the arbitrary object orientations inherent to aerial views, oriented bounding boxes (OBBs) are essential for accurately representing targets, reducing inter-object and inter-frame ambiguity and enhancing performance of object association. To this end, we adapt a multi-stage annotation pipeline to guarantee geometric precision of OBBs.

Most existing MOT algorithms are designed for RGB inputs with axis-aligned boxes, limiting their effectiveness on multispectral and orientation-aware tasks. To overcome this, we further propose a unified adaptation scheme that enables mainstream MOT frameworks to exploit spectral information and OBB annotations. This includes a lightweight Spectral 3D-Stem for spectral-spatial feature extraction compatible with RGB-pretrained weights, an orientation-aware Kalman filter for motion modeling, and an orientation-adaptive transformer framework.

The proposed dataset and adaptation scheme jointly establish a strong foundation for advancing multispectral drone-based multi-object tracking. Extensive experiments and benchmark demonstrate consistent improvements over RGB-based counterparts. Spectral information significantly enhances detection and identity association, particularly for small objects with limited spatial cues. Together, the dataset and methods offer both critical data support and practical modeling strategies, paving the way for future research in orientation-aware, multispectral MOT.

Our principal contributions include: (i) MMOT, the first challenging benchmark for drone-based multispectral dataset multi-object tracking with precise oriented bounding box annotations; (ii) A comprehensive orientation-aware multispectral MOT solution, incorporating the proposed Spectral 3D-Stem module, an orientation-aware Kalman filter, and an end-to-end orientation-aware tracking framework. (iii) A comprehensive benchmark through extensive experimental evaluation, serving as a foundation for future research. All datasets and code are released for public access to facilitate further development and reproducibility.

## 2 Related Work

**Drone-based Multi-Object Tracking Datasets.** The growing interest in MOT from unmanned aerial vehicles has spurred the introduction of specialized datasets tailored to aerial perspectives. The UAVDT dataset[4] specifically targets vehicle detection and tracking, covering a variety of realistic traffic scenarios with annotations of critical attributes such as weather conditions, camera altitude, and viewing angles. Similarly, the VisDrone dataset[5] offers a comprehensive benchmark collected by DJI drones across 14 cities in China, capturing diverse urban and suburban environments, varying illumination, and complex weather conditions. Expanding UAV tracking into wildlife monitoring, the BuckTales dataset[6] provides annotated videos for tracking and re-identifying blackbuck antelopes, presenting unique challenges associated with animal tracking in natural environments.

**Multispectral Datasets for Visual Tracking.** Several MSI datasets have recently been introduced for visual tracking. The HOT dataset [7] includes 50 sequences collected with mosaic snapshot cameras, emphasizing the benefits of spectral diversity in challenging scenarios. Further advancement was driven by the HOTC 2024 challenge, featuring 346 videos captured by various sensors. For drone-based applications, the MUST dataset [8] provides 250 single-object tracking sequences recorded across eight bands under diverse conditions, validating the benefits of spectral data in aerial settings. Despite this progress, these efforts are limited to single-object or general tracking.

**Generic Multi-object Tracking Datasets.** To support diverse tracking scenarios, various generic MOT datasets have been developed. MOTChallenge benchmarks such as MOT15 [9], MOT17 [10], and MOT20 [11], as well as DanceTrack [12] and SportsMOT [13], primarily focus on pedestrian tracking under crowded or low-discriminability conditions. TAO [14] extends this to large-scale, multi-category object tracking, enabling research on category-agnostic models. In the autonomous driving domain, KITTI [15] and BDD100K [16] provide vehicle-centric multi-object tracking datasets collected from vehicle-mounted sensors.

## 3 MMOT Dataset

### 3.1 Construction Principle

The objective of MMOT is to establish a comprehensive and challenging benchmark tailored for drone-based multi-object tracking in real-world scenarios, with a specific focus on integrating rich spectral modalities and precise geometric annotations. To this end, the following principles guided the design and construction of the MMOT dataset:

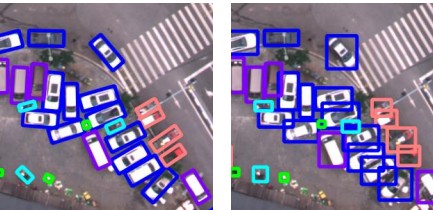

Orientated Bboxes      Axis-aligned Bboxes

Figure 2: Comparison of different annotations in UAV views.

• **Scalable and Diverse Data Foundation.** A fundamental principle in constructing MMOT is ensuring sufficient data volume to support deep model training and reliable evaluation. To this end, we target a large-scale dataset comprising over 100 video sequences and about 500K annotated instances, enabling robust learning across varied conditions and object categories.

• **Broad Coverage of Real-World Challenges and Scenarios.** We target diverse UAV scenes spanning urban, rural, and dynamic environments, with rich variations in object scale, density, occlusion, and camera motion, to comprehensively reflect real-world tracking complexity.

Table 1: Comparison of representative datasets. [†]: The statistics are computed based on publicly available labels.

| Dataset | Scenario | Videos | Total Frames | Total Duration | Total Ann. | Avg. Ann. | Num. of Cat. | Num. of Channels | Oriented Bbox |
|---|---|---|---|---|---|---|---|---|---|
| MOT20[11] | Surveillance | 8 | 13.4K | 535s | 2.1M | 156.7 | 1 | 3 | ✗ |
| DanceTrack[12] | Dancing | 100 | 105.8K | 5292s | – | – | 1 | 3 | ✗ |
| SportsMOT[13] | Sports | 240 | 150.3K | 6015s | 1.6M | 10.8 | 1 | 3 | ✗ |
| UAVDT-MOT[4] | UAV | 50 | 40.4K | 1346s | 763.8K | 18.9 | 3 | 3 | ✗ |
| VisDrone-MOT[†][5] | UAV | 80 | 33.6K | – | 1.1M | 33.6 | 5 | 3 | ✗ |
| **MMOT** (Ours) | UAV | 125 | 13.8K | 2767s | 488.3K | 35.2 | **8** | **8** | ✓ |

• **OBB Annotation for UAV Views.** To address the spatial distortion and reduce inter-frame and inter-object ambiguity as shown in Fig. 2, we adopt OBBs that better conform to object geometry, reduce inter-object error, and improve localization accuracy as well as target association.

## 3.2 Dataset Overview

MMOT is the first large-scale drone-based multispectral MOT dataset, designed to advance research on MOT in challenging aerial scenarios, comprising 125 video sequences and 488.8K annotated OBBs. The category hierarchy is well-structured, comprising three superclasses—*HUMAN* (*pedestrian*), *VEHICLE* (*car*, *van*, *truck*, *bus*), and *BICYCLE* (*bike*, *awning-bike*, *tricycle*)—spanning a total of eight fine-grained object types.

Table 1 summarizes a comparative overview of MMOT and representative generic and drone-based MOT datasets. MOT20, DanceTrack and SportsMOT focus exclusively on pedestrian tracking in constrained settings such as surveillance, group dancing, or sports courts. While these datasets offer large scale and dense annotations, they lack diversity in object types and viewing conditions, and provide only RGB imagery—limiting their utility in modeling the complex motion dynamics and visual degradations typical in drone-based multi-object tracking. Compared with UAVDT and VisDrone, MMOT offers significantly extended tracking durations and higher annotation density, with an average of 35.2 objects per frame. It also supports a broader range of object classes (8 vs. 3 and 5), better reflecting the complexity of real-world UAV deployments involving multi-category and densely packed targets. Most notably, MMOT is the only one among the six datasets that provides both multispectral imagery and precise oriented bounding box annotations, enabling research into multispectral and orientation-aware tracking models.

## 3.3 Dataset Construction

**Data Acquisition.** MMOT was constructed using a UAV equipped with a downward-facing multispectral camera that captures eight spectral bands ranging from the visible to near-infrared spectrum, with data acquired during flights conducted at dynamic altitudes between 80 and 200 meters. To ensure the dataset reflects realistic deployment conditions, data were collected under various weather scenarios, including clear skies, cloudy days, and dense fog.

Meanwhile, a wide range of environments was covered, including urban streets, rural fields, traffic intersections, transit hubs, playgrounds, and sports courts. All frames were precisely registered to ensure pixel-level alignment across spectral channels, then uniformly cropped to $1200 \times 900$ pixels, yielding high-quality multispectral sequences for reliable aerial tracking.

**Annotation.** MMOT is a meticulously curated dataset featuring over 5,000 human-hours of manual annotation, tailored for training, evaluating, and visualizing orientation-aware MOT models in aerial scenarios. It adheres to a strict labeling protocol and integrates enhanced tooling support to ensure both annotation quality and operational scalability. Fig. 3 shows the fine-grained alignment and the challenges of precisely labeling small objects.

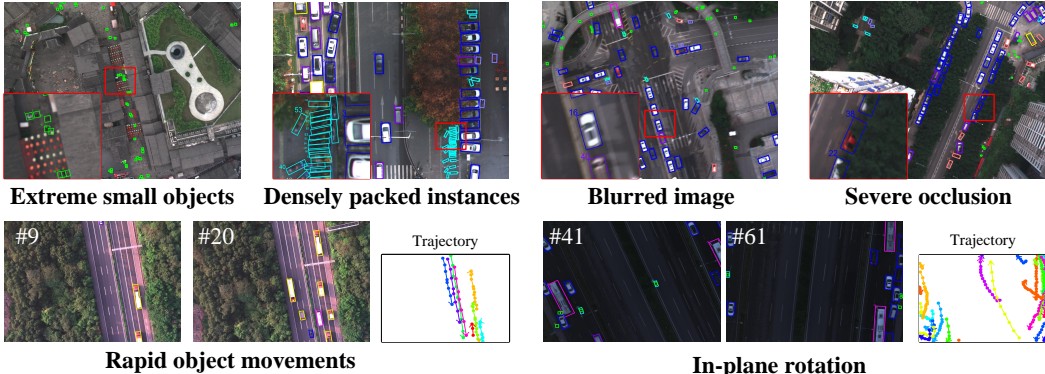

**Extreme small objects**  **Densely packed instances**  **Blurred image**  **Severe occlusion**

**Rapid object movements**  **In-plane rotation**

Figure 3: Example annotations from MMOT showcasing diverse and challenging scenarios. In these scenes, where spatial features are limited due to small object size, clutter or blur, spectral cues provide critical complementary information for reliable discrimination. Zoom in for better visualization.

To achieve high labeling precision and temporal consistency, MMOT assigns each object a unique identity across frames and adopts OBBs. Annotators then follow a five-fold protocol to guarantee annotation quality and exhaustiveness:

- **Exhaustive category coverage.** All instances from predefined categories must be annotated, regardless of size or duration.

- **Spectral assistance.** When a target is not sufficiently discernible in the pseudo-color image, annotators examine other spectral channels to identify the channel in which the target is most distinguishable and use it to determine the target's existence, spatial position, and boundaries.

- **Temporal validation for ambiguous cases.** For objects that are difficult to confirm based on a single frame, annotators are required to review the entire video sequence to determine identity and ensure temporally consistent and accurate annotations.

- **Spatial completeness.** Full object extents must be labeled, even under occlusion, truncation, or motion blur, using temporal context and shape priors.

- **Identity consistency.** Each object must retain a unique ID throughout the video without reassignment or duplication.

Building upon these annotation principles, a multi-stage annotation workflow—consisting of initial box placement, box refinement, identity assignment, identity correction, and expert-level cross-validation—ensures annotation accuracy while supporting large-scale deployment. Over 20 trained annotators handled the main stages, with final review by three senior experts. This comprehensive framework significantly improves annotation efficiency and reliability, providing high-quality labels well-suited for robust multispectral aerial tracking research.

To maintain compatibility with modern MOT models, automatic post-processing is applied. Instances are discarded if their center lies outside the image frame, their intersection-over-foreground (IoF) is less than 0.5, or their bounding box exceeds the image boundary by more than 100 pixels. Objects partially cut by the image boundary but not meeting these removal criteria are retained and labeled as *truncated*.

**Dataset Splitting.** MMOT is partitioned into training and test sets to support robust algorithm development and evaluation under diverse real-world UAV tracking conditions. To ensure fairness and generalization, environmental factors such as lighting conditions and weather states are evenly distributed across the two subsets, and no geographic location or specific scene instance appears in both splits to avoid overfitting. As shown in Fig. 4(a), the final split comprises 75 training sequences and 50 test sequences. The training set contains 8,372 frames, 6,101 identity-consistent tracks, and 292K rotated bounding boxes, while the test set consists of 5,446 frames, 4,527 tracks, and 196K bounding boxes. This careful partitioning avoids distributional bias and ensures that evaluation reflects true generalization to novel spatial and contextual scenarios.

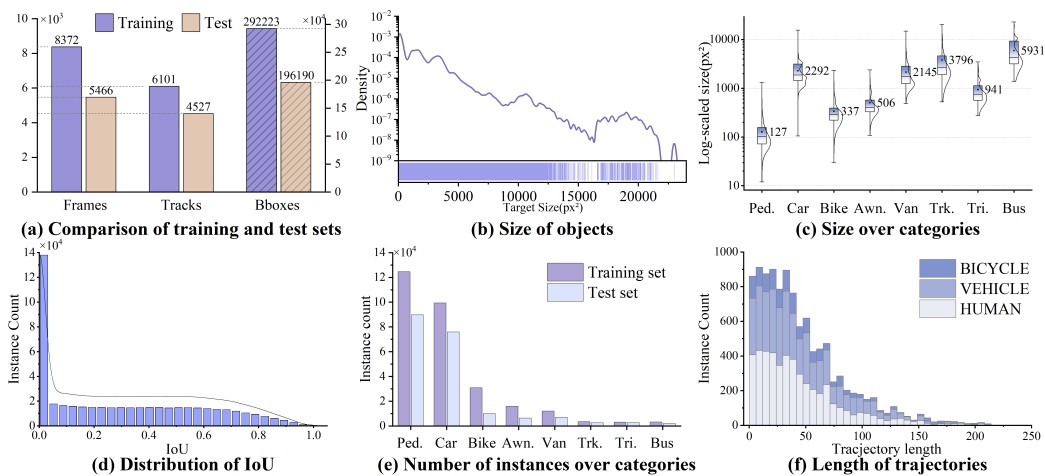

Figure 4: Distributions of dataset split, object size, inter-frame IoU of the same object, instances over object categories, and trajectory lengths in MMOT.

Table 2: Density and motion statistics across datasets. Max: maximum objects per frame; @300px: average number of objects within a 300-pixel radius. Displacement is decomposed into drone (platform motion), object (ground-relative motion after compensation), and total (combined apparent motion). IoU is reported for both object and total motion.

| Dataset | Max ↑ | @300px ↑ | Displacement (pixels)↑ | | | IoU ↓ | |
| | | | Drone | Object | Total | Object | Total |
|---|---|---|---|---|---|---|---|
| VisDrone-MOT | 147 | 14.7 | 2.3 | 2.8 | 4.2 | 0.88 | 0.85 |
| UAVDT-MOT | 82 | 18.0 | 1.4 | 1.1 | 1.2 | 0.91 | 0.91 |
| **MMOT (Ours)** | **155** | **19.4** | **14.1** | **4.3** | **14.4** | **0.68** | **0.30** |

## 3.4 Statistical Analysis

**Size and Density Challenges.** As shown in Fig. 4(b), small objects dominate the overall distribution, highlighting the prevalence of tiny targets. In addition, as shown in Fig. 4(c), all classes demonstrate a wide variance in target size, which reflects the variability in UAV flight altitude and ground sampling distance during data acquisition. Beyond object size, Tab. 2 compares MMOT with existing UAV-based MOT benchmarks in terms of spatial density and motion complexity. MMOT achieves the highest object density, with a maximum of 155 objects per frame and an average of 19.4 targets within a 300-pixel radius, surpassing VisDrone-MOT (147, 14.7) and UAVDT-MOT (82, 18.0). These results emphasize inherent difficulty of MMOT: the predominance of small, low-resolution targets combined with highly variable and locally concentrated densities limit the effectiveness of purely spatial features.

**Inter-frame Displacement and Overlap Analysis.** Inter-frame object dynamics represent a crucial characteristic, as many MOT algorithms rely heavily on consistent motion patterns to maintain identity associations. In drone-to-ground scenarios, the apparent motion of each target arises from two coupled sources: the ego-motion of the drone platform and the intrinsic motion of the object itself. As detailed in Tab. 2, we estimate platform motion via KLT optical flow [17] and decouple it from object motion to evaluate both components independently. Compared with VisDrone-MOT (drone/object/total displacements of 2.3/2.8/4.2 pixels) and UAVDT-MOT (1.4/1.1/1.2 pixels), MMOT exhibits substantially larger dynamics, with average drone-, object-, and total-displacement magnitudes of 14.1, 4.3, and 14.4 pixels, respectively. This strong apparent motion is accompanied by a markedly lower inter-frame IoU, averaging 0.68 for object motion and only 0.30 for total motion—far below the 0.9 range observed in previous datasets. The IoU distribution in Fig. 4(d) further supports this finding, showing that most objects retain overlaps below 0.1, a condition rarely seen in conventional MOT scenarios. These results highlight the difficulty of achieving reliable inter-frame associations using

motion cues alone, as the combined effects of small object size, strong ego-motion, and rapid local movements severely disrupt spatial continuity across frames.

**Long-tail Property of Class Distribution and Trajectory Duration.** As shown in Fig. 4(e) and Fig. 4(f), we analyze the class-wise instance quantity distribution and the trajectory duration distribution, and both distributions exhibit a pronounced long-tailed behavior. This long-tailed distribution reflects a natural bias toward frequently observed small objects, such as pedestrians and cars, as well as short-lived tracks caused by fast motion. Such imbalances in object classes and durations present key challenges for real-world MOT algorithms.

## 4 Multispectral and Orientation-Aware MOT Scheme

To address the limitations of existing MOT algorithms in handling multispectral inputs and leveraging precise OBB annotations, we propose a unified Multispectral and Orientation-Aware MOT Scheme. Following this design, we adapt eight representative MOT algorithms *SORT* [18], *ByteTrack* [19], *OC-SORT* [20], *BoT-SORT*[21], *MOTR* [22], *MOTRv2* [23], *MeMOTR* [24] and *MOTIP* [25] as well as a detection algorithm *YOLOv11* [26].

### 4.1 Spectral 3D-Stem for Multispectral Tracking

**Channel Mismatch in Multispectral Tracking.** Conventional RGB-based tracking models are designed to process images $I_{\text{RGB}} \in \mathbb{R}^{H \times W \times 3}$, whereas multispectral imagery provides input $I_{\text{MSI}} \in \mathbb{R}^{H \times W \times 8}$. This mismatch in channel dimensions renders direct application of pretrained CNNs infeasible. A naive solution is to replace the first convolutional layer to accept 8-channel. This design forces direct compression of spectral features through a single convolution layer, limiting expressive capacity. Moreover, it breaks compatibility with widely used RGB-pretrained weights, hindering transfer learning and requiring re-initialization, hurting training stability.

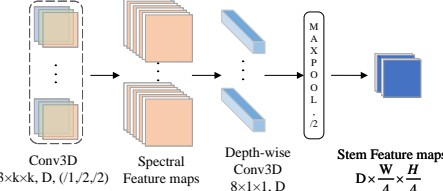

Figure 5: Our proposed Spectral 3D-Stem module employs a Conv3D to extract spectral-spatial features, followed by a depthwise Conv3D to fold the spectral dimension.

**Spectral-Spatial Feature Encoding via Spectral 3D-Stem.** We propose a lightweight yet effective Spectral 3D-Stem module for joint spectral–spatial feature extraction. As illustrated in Fig. 5, a 3D convolution with a spectral kernel size of 3 slides along the spectral axis to capture local spectral variations and produce eight groups of feature maps, each corresponding to a specific spectral band. Subsequently, another 3D convolution with a spectral kernel size of 8 aggregates information across the entire spectral range while preserving the learned spatial semantics.

**Efficient Parameter Reuse with Minimal Overhead.** Our design ensures that the Conv3D layer maintains the same number of learnable parameters as its RGB counterpart, enabling seamless reuse of pretrained RGB weights. Specifically, the added depthwise Conv3D introduces only $8 \times D$ extra parameters, where $D$ is the output channel dimension. This architectural alignment allows initialization from well-trained RGB weights, facilitating stable convergence and efficient optimization without compromising the model's capacity to capture multispectral cues.

### 4.2 Tracking-by-Detection with Oriented State Estimation

We extend the original Kalman filter based motion model used by detection-based trackers to incorporate orientation explicitly. Specifically, an orientation-aware motion state is introduced as: $\mathbf{x} = [u, v, s_1, s_2, \theta, \dot{x}, \dot{y}, \dot{s}_1, \dot{s}_2, \dot{\theta}]^\top$, where $(u, v)$ represents the oriented bounding box center coordinates, $s_1$ and $s_2$ denote size parameters whose definitions vary across methods, $\theta$ denotes the orientation angle, and $\dot{x}, \dot{y}, \dot{s}_1, \dot{s}_2, \dot{\theta}$ represent the corresponding velocities. For data association, we replace the IoU computation with an orientation-aware IoU(rIoU) metric, accurately capturing spatial relationships between oriented bounding boxes.

### 4.3 Orientation-Sensitive Architectures for End-to-End Tracking

**Angle Prediction Head.** Tracking-by-query methods simultaneously predict object locations and identities in an end-to-end manner, typically built upon DETR-like architectures [27] and their variants. To enable these methods to handle oriented bounding boxes, we introduce an additional *angle head* branch parallel to the *box head*, explicitly predicting a normalized orientation angle $\hat{\theta} \in [0, 1]$. Given a decoder embedding $x$, the predicted oriented bounding box is obtained as:

$$([\hat{x}, \hat{y}, \hat{w}, \hat{h}], \hat{\theta}) = \sigma(\text{FFN}_{\text{box}}(x), \text{FFN}_{\text{angle}}(x)), \tag{1}$$

where $\sigma(\cdot)$ denotes a sigmoid activation.

**Iterative Angle Refinement.** Similar to Deformable-DETR [28], we refine the predicted angle $\hat{\theta}$ progressively across decoder layers. Given the previous angle prediction $\hat{\theta}_p$, the regression angle $\delta\hat{\theta}$ and the *le135* format, the actual angle $\theta_r$ is computed as: $\theta_r = (\sigma(\sigma^{-1}(\hat{\theta}_p) + \delta\hat{\theta}) - \frac{1}{4}) * \pi$.

**Optimization Objective.** We adopt a similar optimization objective as the original methods, employing L1-loss on the five-dimensional oriented bounding box parameters $(\hat{x}, \hat{y}, \hat{w}, \hat{h}, \hat{\theta})$ and replacing the standard IoU loss with a rIoU loss to encourage accurate regression of oriented bounding boxes.

## 5 Experiments

### 5.1 Experimental Settings

We conduct extensive experiments on the MMOT dataset under two input modalities: RGB and MSI. For RGB-based evaluation, we synthesize pseudo-RGB images by selecting bands 5, 3, and 2 from the MSI cube, which approximately correspond to the RGB spectrum. For MSI-based evaluation, all eight spectral channels are utilized. All models incorporate the proposed Spectral 3D-Stem for effective multispectral feature extraction in MSI-based experiments. For both RGB and MSI settings, all models are adapted to support rotated bounding boxes using the orientation-aware strategies detailed in Sec. 4.2 and Sec. 4.3. Additional hyperparameters are detailed in the appendix.

To provide a comprehensive assessment of tracking algorithms evaluated on MMOT, we follow MOT benchmarks [12, 13], utilizing CLEAR metrics [29], IDF1 [30], and HOTA [31]. Given the multi-class nature of our dataset, we adopt two category-aware aggregation approaches: class-averaged evaluation and detection-averaged evaluation.

Table 3: Comparison of representative MOT algorithms with MSI input on the MMOT dataset.

| Type | Method | Class-Averaged | | | | | Detection-Averaged | | | | |
|---|---|---|---|---|---|---|---|---|---|---|---|
| | | HOTA | MOTA | IDF1 | DetA | AssA | HOTA | MOTA | IDF1 | DetA | AssA |
| Tracking by Detection | SORT [18] | 27.2 | 24.3 | 29.1 | 25.7 | 30.0 | 35.0 | 25.7 | 33.7 | 27.6 | 44.8 |
| | ByteTrack [19] | 40.5 | 34.2 | 44.1 | 37.0 | 46.2 | 46.0 | 37.8 | 46.7 | 41.9 | 51.5 |
| | OC-SORT [20] | 29.5 | 25.1 | 31.9 | 27.3 | 32.8 | 37.5 | 27.5 | 37.0 | 29.5 | 48.0 |
| | BoT-SORT [21] | **53.6** | **46.2** | **61.0** | **45.7** | **64.6** | **60.7** | **59.4** | **69.4** | **55.0** | **68.7** |
| Tracking by Query | MOTR [22] | 39.0 | 26.5 | 44.6 | 27.1 | 60.1 | 48.4 | 32.2 | 54.7 | 35.4 | 68.4 |
| | MOTRv2 [23] | **49.2** | **43.1** | **57.3** | **37.8** | **67.7** | **54.5** | **50.9** | **64.6** | **44.1** | 68.8 |
| | MeMOTR [24] | 42.3 | 31.3 | 45.9 | 29.3 | 66.3 | 50.9 | 40.8 | 56.0 | 37.1 | **70.9** |
| | MOTIP [25] | 39.0 | 28.8 | 43.9 | 33.8 | 49.6 | 43.1 | 37.3 | 46.3 | 43.7 | 43.8 |

### 5.2 Experimental Results and Analysis

**MSI-based Overall Performance.** All methods are evaluated under comprehensive and fair conditions, with detailed results shown in Tab. 4. Among all evaluated trackers, BoT-SORT achieves the best overall performance, reaching class-averaged metrics of 53.6 HOTA, 46.2 MOTA, and 61.0 IDF1, and detection-averaged metrics of 60.7 HOTA, 59.4 MOTA, and 69.4 IDF1. This superior performance benefits significantly from high-quality detection proposals generated by YOLOv11

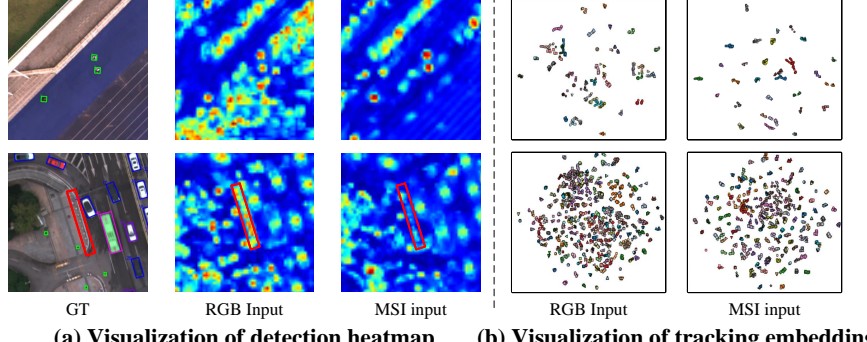

| GT | RGB Input | MSI input | RGB Input | MSI input |

**(a) Visualization of detection heatmap**     **(b) Visualization of tracking embedding**

Figure 6: Visualization of multispectral benefits for detection and tracking. (a) MSI input enhances response for small targets while suppressing background confusion. (b) MSI input leads to more compact and better-separated feature clusters, enhancing discriminability for identity association.

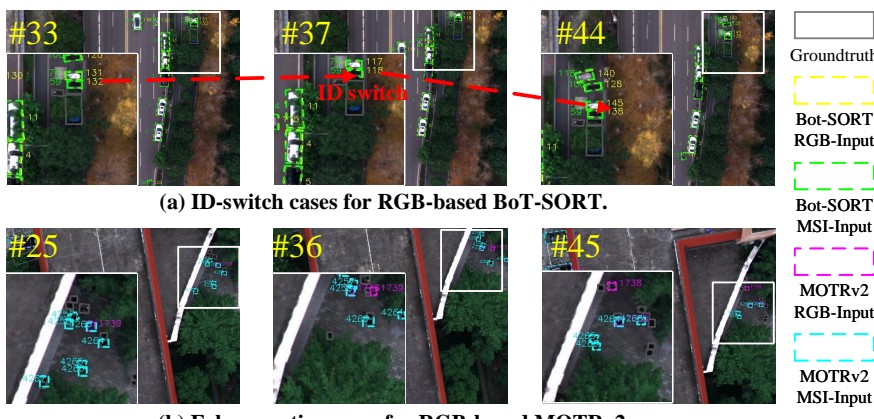

**(a) ID-switch cases for RGB-based BoT-SORT.**

**(b) False negative cases for RGB-based MOTRv2.**

Figure 7: Comparison of two representative trackers (BoT-SORT and MOTRv2) using RGB and MSI inputs. Each tracker is color-coded consistently across scenes. IDs are shown on the right and left side for RGB-based and MSI-based results. To keep brief, IDs of ground-truth are omitted.

and the robust optical-flow module in BoT-SORT, which effectively accounts for camera motion. Similarly benefiting from YOLOv11 detections, MOTRv2 ranks second among all models, achieving class-averaged metrics of 49.2 HOTA, 43.1 MOTA, and 57.3 IDF1, alongside detection-averaged metrics of 54.5 HOTA, 50.9 MOTA, and 64.6 IDF1. Notably, MeMOTR achieves the highest detection-averaged AssA of 70.9, substantially outperforming other methods. This highlights its effectiveness in processing multiple frames, underscoring its advanced capability for multi-frame association in complex tracking scenarios.

**Benefits of Multispectral Cues.** To further quantify the benefits of multispectral input beyond overall performance, we compare the same tracking algorithms under RGB and MSI domains across different superclasses. As shown in Tab. 4, all evaluated models exhibit consistent gains in super-class averaged HOTA scores when leveraging multispectral imagery, underscoring the effectiveness of spectral cues. The performance improvements are particularly prominent in the *HUMAN* category, which features numerous small, low-texture, and densely distributed instances. Specifically, MOTR achieves a +7.0 increase in HOTA, MOTRv2 +7.0 and MeMOTR +7.3.

Table 4: Class-wise HOTA comparison between RGB-based and MSI-based models on MMOT.

| Method | Domain | HUM. | VEH. | BIC. | SuperCls-avg. |
|---|---|---|---|---|---|
| MOTR [22] | RGB | 19.4 | 64.1 | 30.4 | 38.16 |
| | MSI | 26.4 | 66.1 | 32.3 | 41.64 (↑3.48) |
| MOTRv2 [23] | RGB | 29.0 | 67.1 | 37.2 | 44.48 |
| | MSI | 36.4 | 70.1 | 39.0 | 48.53 (↑4.05) |
| MeMOTR [24] | RGB | 24.3 | 63.5 | 34.3 | 40.75 |
| | MSI | 31.6 | 66.8 | 35.6 | 44.69 (↑3.94) |

These results highlight the value of spectral cues in enhancing discriminability under challenging conditions with degraded spatial resolution.

On the other hand, Fig. 6 intuitively illustrates the advantages introduced by multispectral input through detection heatmaps and tracking embeddings. In the top row (left panel), MSI input produces cleaner and more focused heatmaps that sharply localize true targets, whereas RGB-based responses are often diffused or suppressed due to background clutter. Moreover, in the bottom row, MSI effectively suppresses false activations from visually similar distractors or cluttered regions, which remain prominent in the RGB domain. For tracking embeddings (right panel), we visualize identity embeddings via dimensionality reduction, where each color and marker denotes a distinct ID. Compared to RGB, MSI input yields more compact and clearly separated clusters, reflecting improved feature discriminability and reduced identity ambiguity. Collectively, these visualizations highlight how spectral cues offer valuable complementary information for both detection and association under complex aerial conditions.

**Qualitative Comparison of RGB and MSI Inputs.** As shown in Fig. 7, multispectral input leads to improved tracking performance under visually challenging conditions. In the top row, BoT-SORT with RGB input (yellow boxes) exhibits multiple ID switches and missed detections for bike targets in densely populated scenes. In the bottom row, MOTRv2 with MSI input (cyan boxes) exhibits more stable associations and better recall than its RGB counterpart (pink boxes), particularly in tracking multiple small, low-resolution pedestrian instances. Although neither model achieves perfect tracking under extremely small object, the MSI version detects and maintains a significantly higher number of correct tracks, aided by the spectral separability of human targets. These qualitative observations are not isolated cases, but rather representative patterns observed across the dataset. They demonstrate that multispectral input effectively mitigates identity switches, reduces false detections, and enhances overall tracking robustness under challenging conditions.

**Spectral 3D-Stem Analysis.** We further investigate the contribution of the proposed Spectral 3D-Stem by replacing it with a naive 2D-stem baseline. As shown in Tab. 5, the Spectral 3D-Stem consistently improves class-averaged tracking performance across all evaluated models. The most significant gains appear in tracking-by-query frameworks, with HOTA increases of +3.1 for MOTR and +3.8 for MeMOTR. These improvements demonstrate the capability of the Spectral 3D-Stem to effectively capture inter-band correlations and fine-grained spectral–spatial context. In addition, its compatibility with pretrained RGB weights facilitates stable optimization and faster convergence during finetuning. Overall, these results confirm that the Spectral 3D-Stem provides an efficient architectural for multispectral learning, yielding richer representations and more robust performance under challenging tracking scenarios.

Table 5: Ablations on the Spectral 3D-Stem module.

| Method | Stem | HOTA | MOTA | IDF1 |
|---|---|---|---|---|
| ByteTrack [19] | 2D | 40.3 | 35.8 | 43.8 |
|  | 3D | 40.5 | 34.2 | 44.1 |
| BoT-SORT [21] | 2D | 52.8 | 45.4 | 59.2 |
|  | 3D | 53.6 | 46.2 | 61.0 |
| MOTR [22] | 2D | 35.9 | 23.6 | 39.7 |
|  | 3D | 39.0 | 26.5 | 44.6 |
| MeMOTR [24] | 2D | 38.5 | 25.6 | 40.5 |
|  | 3D | 42.3 | 31.3 | 45.9 |

## 6 Conclusion

We introduce MMOT, the first large-scale drone-based multispectral MOT dataset with oriented bounding boxes, featuring 125 videos and 488.8K high-quality OBB annotations across eight object categories. To fully exploit this setting, we propose a unified adaptation scheme that integrates a Spectral 3D-Stem and orientation-aware tracking modules. Extensive experiments on eight representative MOT models demonstrate consistent gains from multispectral input, especially for small and crowded targets. All data and code are released to support further research.

**Limitation.** Annotating high-quality OBBs requires substantial manual effort. Future work will explore scalable annotation and unsupervised learning approaches.

**Acknowledgment.** This work was financially supported by the National Natural Science Foundation of China (No. 62571031) and the Chongqing Excellent Young Scientists Fund (No. CSTB2025NSCQ-JQX0017).

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

# Supplemental Material for
# MMOT: The First Challenging Benchmark for Drone-based Multispectral Multi-Object Tracking

## A   Appendix

In this appendix, we provide additional details, analysis, results, and discussions of the MMOT project including:

### A.1   Category Structure
Overview of the hierarchy of object classes in MMOT, organized into three superclasses and eight fine-grained categories.

### A.2   Annotation Tooling and Pipeline
Description of customized labeling tool designed and the multi-stage annotation process for efficient and accurate OBB-MOT data collection.

### A.3 Visuliazition of Challenging Scenarios in MMOT
Illustrates representative frames across diverse tracking challenges, including single difficulties like small objects and density as well as complex, multi-factor scenarios.

### A.4   Spatial Distribution of Object Centers
Visualization of target center heatmaps across all sequences, reflecting UAV framing patterns and spatial coverage diversity.

### A.5   Sequence Length Statistics
Statistical distribution of video sequence lengths across the dataset to support benchmarking on variable temporal scales.

### A.6   Camera Spectral Configuration
Specific spectral band configuration of the multispectral camera used in MMOT.

### A.7   Spectral 3D-Stem Architecture
Detailed architectural comparison between Spectral 3D-Stem, 2D-Stem, and original ResNet-style stems, with emphasis on compatibility and efficiency.

### A.8   Experimental Implementation Details
Comprehensive setup for all baseline experiments, including model configurations, training schedules, and modality-specific adjustments.

### A.9   Computational Resources
Hardware specifications used in experimentation, with emphasis on GPU setups.

### A.10   Detailed Comparison across Modalities and Stem Variants
Fine-grained evaluation of MOT performance under RGB vs. MSI modalities and different stem designs (2D vs. 3D).

### A.11   Impact of Detector Quality on Tracking Performance
Analysis of how different detection (YOLOv11-L vs. Deformable-DETR) affect downstream tracking-by-detection accuracy.

### A.12   Broader Societal Impacts
Discussion of the societal implications of multispectral tracking, covering both beneficial applications and potential risks.

### A.13   Licenses for Existing Assets
Licensing terms and usage acknowledgments for all third-party datasets and codebases integrated in this work.

## A.1  Category Structure

The category hierarchy in our dataset is systematically organized into three superclasses and eight classes. The superclasses include HUMAN, VEHICLE, and BICYCLE. The HUMAN superclass contains *pedestrian*; the VEHICLE superclass includes *car*, *van*, *truck*, and *bus*; while the BICYCLE superclass comprises *tricycle*, *bike*, and *awning-bike*. For clarity and consistency, each fine-grained category is assigned a standardized abbreviation throughout the dataset: Ped. (Pedestrian), Car, Van, Tru. (Truck), Bus, Tri. (Tricycle), Bike, and Awn. (Awning-bike). The example of each class is shown in Fig. 8.

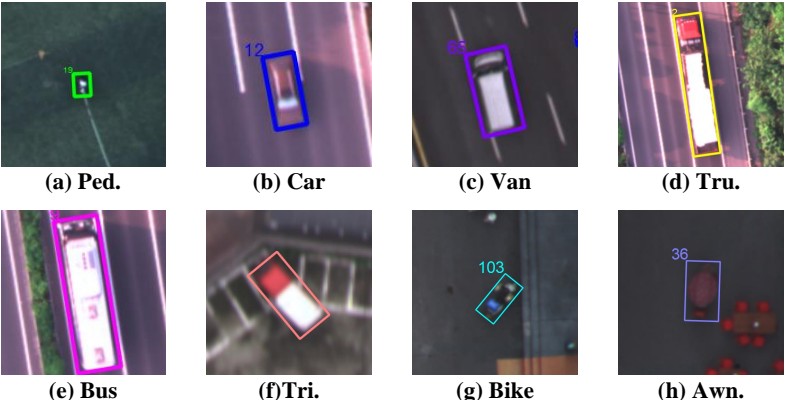

| (a) Ped. | (b) Car | (c) Van | (d) Tru. |

| (e) Bus | (f)Tri. | (g) Bike | (h) Awn. |

Figure 8: Category examples in the MMOT dataset. Each fine-grained class is shown with its corresponding bounding box and abbreviation: (a) Pedestrian (Ped.), (b) Car, (c) Van, (d) Truck (Tru.), (e) Bus, (f) Tricycle (Tri.), (g) Bike, and (h) Awning-bike (Awn.).

## A.2  Annotation Tooling and Pipeline

MMOT is a meticulously curated dataset featuring over 5,000 human-hours of manual annotation, tailored for training, evaluating, and visualizing rotation-aware MOT models in aerial scenarios. It adheres to a strict labeling protocol and integrates enhanced tooling support to ensure both annotation quality and operational scalability.

**Enhanced Tool Support for OBB-MOT.** To facilitate the annotation of oriented bounding boxes (OBBs) in multi-object tracking tasks, we developed a dedicated labeling tool based on *X-AnyLabeling* [32], enhanced with several key features tailored for MMOT annotation. The interface displays each annotated object with a color corresponding to its category and the object ID is shown at the top-left corner of its bounding box in the same color, aiding intuitive identity tracking across frames as shown in Fig. 9(a).

- **Real-time Tracking Assistance.** The tool supports real-time tracking assistance through automated frame-to-frame ID association and interactive prompts. Specifically, four types of label status are detected as shown in Fig. 9(b)(c)(d)(e): (i) duplicate IDs within the same frame, (ii) ID-category mismatch across adjacent frames, (iii) IDs that disappear from the previous frame, and (iv) newly introduced IDs. The first two cases are categorized as error-level issues and the latter two as warnings. All alerts are presented in the warning panel on the right side of the interface as shown in Fig. 9(a). In addition, the corresponding object IDs within the annotation view are recolored: red for error-level issues and white for warnings. These indicators override the default category color scheme, enabling annotators to quickly identify and correct label inconsistencies.

  To better assist identity verification, users can optionally overlay the previous frame's annotations as semi-transparent gray boxes. Additionally, trajectory lines between adjacent frames help clarify object motion and support temporally coherent labeling. Warnings and errors are also reflected in the overlaid annotations from previous frames, assisting annotators in identifying temporal inconsistencies.

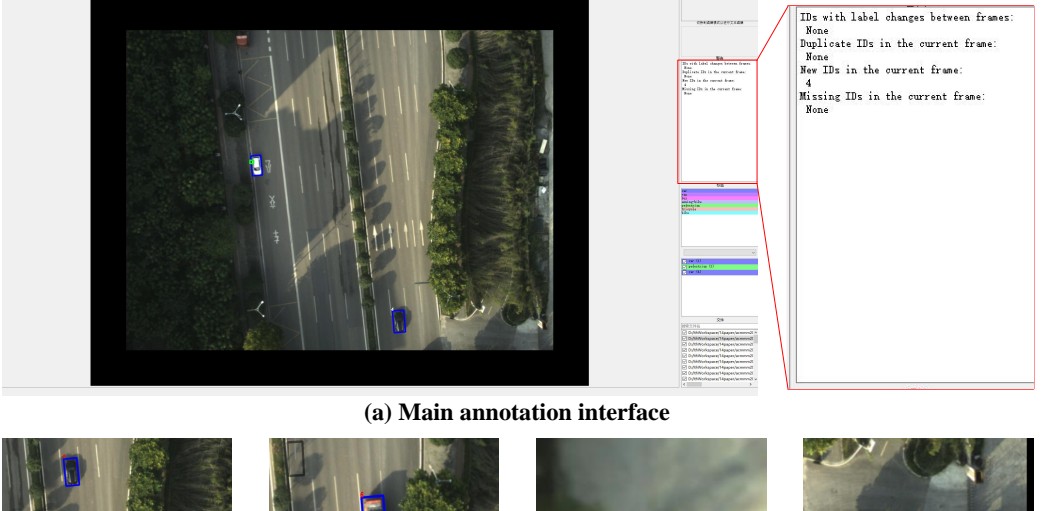

**(a) Main annotation interface**

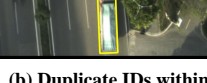

**(b) Duplicate IDs within the same frame**

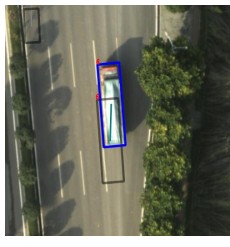

**(c) ID-category mismatch across adjacent frames**

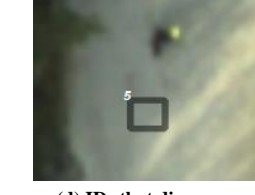

**(d) IDs that disappear from the previous frame**

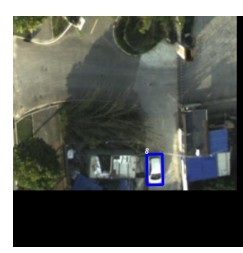

**(e) Newly introduced IDs**

Figure 9: Interface and functionalities of the customized annotation tool for OBB-MOT. (a) shows the main annotation interface, where bounding boxes are color-coded by class and ID labels are rendered in the same color at the top-left corner of each box. Right panel of (a) displays the error/warning panel.

- **Batch ID Operations.** For efficient label management, batch operations are supported. Annotators can batch-replace specific IDs or interchange IDs across frames to correct misassignments. These utilities reduce redundant manual operations in long sequences.

- **Format Support for OBB-MOT.** The tool also includes format conversion utilities, supporting bidirectional conversion between the custom OBB annotation format and both MOT and YOLO-style formats. This allows seamless integration with popular tracking and detection pipelines.

Overall, this tool provides critical support for large-scale, high-quality OBB-MOT annotation, enabling consistent identity management and reducing human error across densely populated aerial scenes.

**Scalable Multi-Stage Pipeline.** As shown in Tab. 6, a five-stage annotation pipeline—consisting of initial box placement, box refinement, identity assignment, identity correction, and expert-level cross-validation—ensures annotation accuracy while supporting large-scale deployment. Over 20 trained annotators handled the main stages, with final review by three senior experts.

**Model-Ready Post-processing.** To satisfy the principle of spatial completeness—requiring full object extents to be labeled even under occlusion or truncation—we pad each image with a 200-pixel-wide black margin on all sides during annotation. This ensures that objects partially leaving the field of view can still be fully enclosed by OBBs, with complete geometry preserved. Note that the 200-pixel padding is applied only during annotation to ensure spatial completeness and is removed prior to model training and evaluation.

To enhance compatibility with modern MOT algorithms, automatic post-processing is performed after annotation. Specifically, any instance is discarded if: (i) its center falls outside the original image region, (ii) its intersection-over-foreground (IoF) is less than 0.5, or (iii) its OBB extends more than 100 pixels beyond the original image boundary. Objects that partially lie outside the original

frame are retained and marked as *truncated*, supporting evaluation protocols that account for visibility constraints.

These extensions significantly improve annotation efficiency and reliability, providing high-quality labels well-suited for robust multispectral aerial tracking research.

Table 6: Five-Stage Annotation Pipeline for MMOT Dataset

| Annotation Stage | Annotation Description |
|---|---|
| **Stage 1** 
 **Initial Box Placement** | A detection model automatically generates coarse oriented bounding box proposals for all visible object instances across eight predefined categories. Annotators then verify and adjust these boxes, ensuring they tightly fit object geometry and orientation. For partially occluded objects, annotators infer complete regions using temporal context and shape priors. |
| **Stage 2** 
 **Box Refinement** | Every bounding box is reviewed for geometric accuracy. Annotators refine imprecise OBBs by verifying if they represent the object's minimum enclosing box with correct orientation. Temporal consistency is cross-validated to confirm detection stability across frames, especially in complex transitions (e.g., objects entering or exiting the field of view, becoming fully occluded, or undergoing temporary disappearance). |
| **Stage 3** 
 **Identity Assignment** | An automatic identity initialization is conducted using an IoU-based frame-to-frame association strategy. Annotators review and adjust identity continuity with special attention to: i) correct initialization and termination of new IDs, ii) identity switches due to occlusion or motion, and iii) consistent labeling in dense clusters where tracking ambiguity is high. Category consistency is also checked throughout ID lifespan. |
| **Stage 4** 
 **Identity Correction** | Annotators conduct identity-level inspection. Each track is examined to ensure its temporal coherence and semantic correctness. Cases of identity loss, switch, or fragmentation are manually corrected. For every ID initialization and disappearance, annotators must verify the reason (e.g., occlusion, entering/exiting view) and mark it accordingly. |
| **Stage 5** 
 **Expert-level** 
 **Cross-Validation** | Final validation is conducted by three senior annotators. Each video is randomly assigned to a second reviewer (not involved in its initial labeling). Annotators cross-check OBB quality, ID consistency, and temporal coverage. Disagreements or ambiguous regions are flagged and jointly reviewed. Each final label must pass agreement from at least two experts. Disputed samples undergo iterative refinement until consensus is reached. |

## A.3 Visuliazition of Challenging Scenarios in MMOT

Figure 10 showcases a wide range of visual challenges captured in the MMOT dataset, progressing from isolated difficulties as well as complex combinations. These include small targets immersed in visually cluttered environments, densely packed objects that create spatial ambiguity, and highly structured yet chaotic urban intersections. Additionally, the dataset captures platform jitter during flight, which distorts spatial consistency, and platform rotation, which causes sudden viewpoint shifts. Temporal dynamics further complicate tracking, as shown in rapid object motion and composite motion patterns involving multiple simultaneous challenges. These examples reflect the diversity and complexity of real deployment conditions. In such settings, multispectral imagery provides a valuable complement to RGB input, offering additional cues that enhance contrast between objects and background and improve robustness against motion blur, occlusion, and appearance variation.

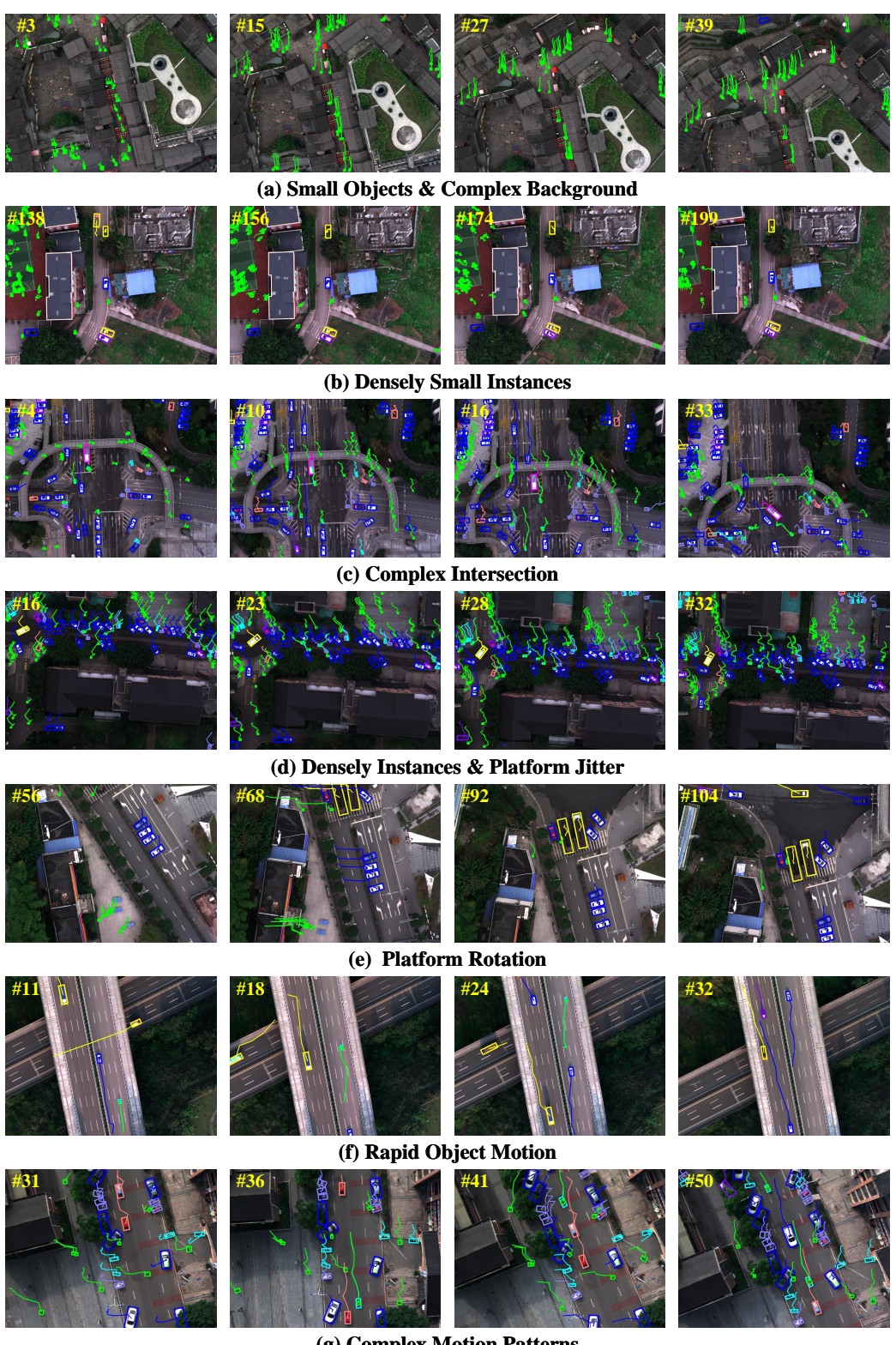

Figure 10: Illustration of representative and challenging tracking scenarios in MMOT. These real-world situations feature dense small target, complex background, platform jitter and rotation, rapid object motion and complex motion patterns. Multispectral sensing provides additional spectral cues beyond RGB, offering more robust solutions under such complex and noisy conditions.

## A.4 Spatial Distribution of Object Centers

Figure 11 illustrates the density heatmap of all object center locations aggregated over the entire MMOT dataset. The spatial distribution reflects typical UAV imaging behavior, with a tendency to track objects near the center of the frame. However, substantial dispersion across the entire image plane can also be observed, indicating that targets appear under unconstrained and diverse viewpoints. This broad spatial coverage highlights the complexity of the dataset and the necessity for detection and tracking models to remain robust across varying object positions.

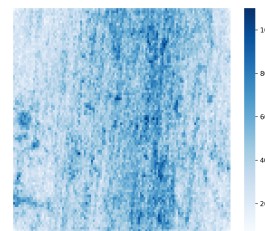

Figure 11: Heatmap visualization of object center distributions across the MMOT dataset.

## A.5 Sequence Length Statistics

Figure 12 presents the sequence length distribution in MMOT. The dataset comprises a total of 125 sequences, split into 75 for training and 50 for testing. Panels (a) and (b) visualize the training set in two parts for clarity, while panel (c) illustrates the test set.

The sequence lengths vary significantly, with the shortest clips containing fewer than 50 frames and the longest exceeding 470 frames. This variability mirrors the natural inconsistencies in UAV video durations under real-world constraints, such as battery limits, scene dynamics, or operational interruptions. The inclusion of such a wide range supports the development and evaluation of models under varying temporal contexts.

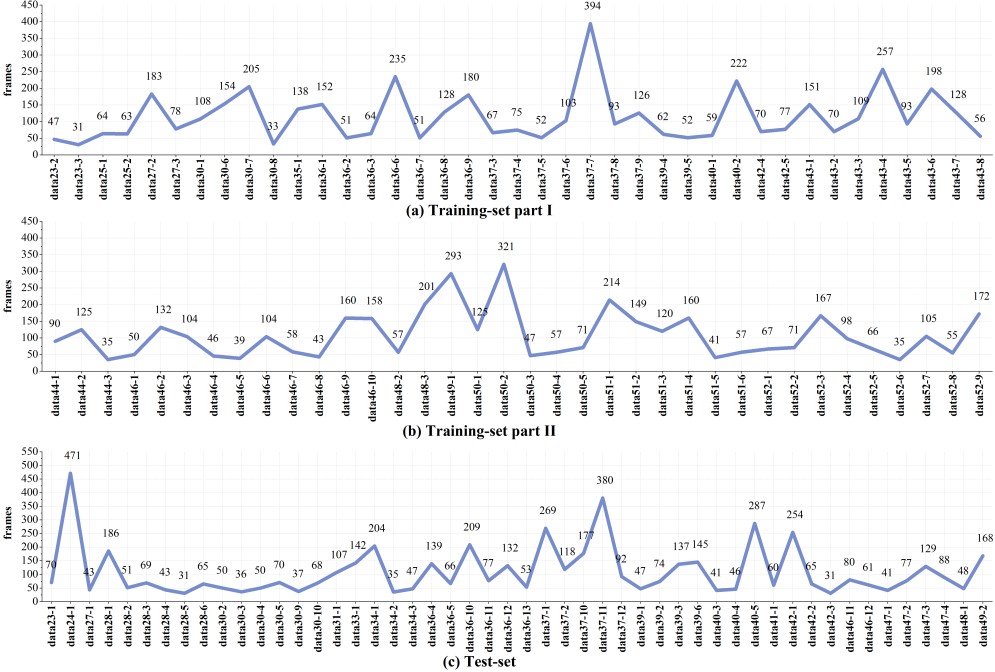

Figure 12: Frame count distribution across sequences in the MMOT dataset. (a) and (b) show the first and second halves of the training set, respectively; (c) shows the full test set.

## A.6 Camera Spectral Configuration

The MMOT dataset was collected using an eight-band multispectral camera covering the visible to near-infrared range (395–950 nm). Table 7 summarizes the detailed spectral configuration. Among

these, bands 5 (660.0 nm), 3 (550.0 nm), and 2 (487.5 nm) are selected as the RGB proxy channels because their center wavelengths closely match the canonical RGB centers. This alignment provides a physically meaningful mapping for RGB visualization and ensures compatibility with RGB-based detection and tracking models.

Table 7: Spectral configuration of the eight-band multispectral camera used for MMOT data acquisition.

| Band | Start (nm) | End (nm) | Center (nm) | Nominal Color |
|------|-----------|----------|-------------|---------------|
| 1 | 395 | 450 | 422.5 | Violet |
| 2 | 455 | 520 | 487.5 | Blue |
| 3 | 525 | 575 | 550.0 | Green |
| 4 | 580 | 625 | 602.5 | Orange |
| 5 | 630 | 690 | 660.0 | Red |
| 6 | 705 | 745 | 725.0 | Red Edge |
| 7 | 750 | 820 | 785.0 | NIR1 |
| 8 | 825 | 950 | 887.2 | NIR2 |

## A.7 Spectral 3D-Stem Architecture

To better illustrate the architectural differences among input stem designs, we summarize key configurations in Table 8. The original ResNet-style stem accepts 3-channel RGB input and applies a single 2D convolution. A naive 2D-stem extension increases the input channels from 3 to 8 but retains purely spatial convolutions, failing to exploit the spectral structure.

In contrast, our proposed **Spectral 3D-Stem** restructures the input into a $1{\times}8$-channel tensor and employs a 3D convolution to jointly capture spectral and spatial patterns. To project back to a spatial feature map, we introduce a depth-wise 3D convolution for per-channel spectral folding, effectively collapsing the spectral axis. This spectral folding module introduces only 512 additional parameters $(8{\times}64)$, while the initial 3D convolution is designed to match the parameter layout of the original 2D stem $(3{\times}64{\times}7{\times}7)$. This enables direct reuse of pretrained RGB weights in the first convolutional layer, ensuring strong initialization and efficient convergence.

Overall, the Spectral 3D-Stem balances spectral modeling capability with parameter efficiency and transferability, making it highly practical for adapting RGB-pretrained networks to multispectral inputs.

## A.8 Experimental Implementation Details

We evaluate and benchmark eight representative MOT algorithms from two mainstream paradigms: (i)tracking-by-detection methods including *SORT* [18], *ByteTrack* [19], *OC-SORT* [20] and *BoT-SORT*[21]; and (ii)tracking-by-query methods including *MOTR* [22], *MOTRv2* [23], *MeMOTR* [24] and *MOTIP* [25].

**Tracking-by-Detection Methods.** To ensure a fair comparison across all tracking-by-detection baselines, YOLOv11-L-OBB is trained for OBB prediction. For the RGB modality, we utilize the official OBB-enabled version of YOLOv11-L [26]. For the MSI modality, we replace its input stem with the proposed Spectral 3D-Stem (with kernel size $k=3$ and output dimension $D=64$) to accommodate 8-channel input and extract spectral-spatial features effectively.

Both detectors are trained on the MMOT training set using input images resized to $960{\times}1280$. In the RGB domain, we follow the default learning rate and optimization schedule from the original YOLOv11-L implementation. In the MSI domain, the learning rate for the Spectral 3D-Stem module is scaled by a factor of 10 to, while other parameters retain the default learning rate. All models are trained for 40 epochs.

All trackers rely solely on motion cues for association without employing appearance-based ReID modules, thus eliminating the need for additional training. A rotation-aware Kalman filter is implemented and used to replace the original axis-aligned version, enabling precise state estimation for

Table 8: Comparison of Input/Output Shapes and Parameters among Different Stem Designs

| Component | Original Stem | 2D-Stem (MSI) | Spectral 3D-Stem (MSI) |
|---|---|---|---|
| Input Shape | $3 \times H \times W$ | $8 \times H \times W$ | $1 \times 8 \times H \times W$ |
| First Conv Layer | 2D Conv
$7 \times 7$, stride 2
in_ch=3, out_ch=64 | 2D Conv
$7 \times 7$, stride 2
in_ch=8, out_ch=64 | 3D Conv
$3 \times 7 \times 7$, stride (1,2,2)
in_ch=1, out_ch=64 |
| Intermediate Output | $64 \times \frac{H}{2} \times \frac{W}{2}$ | $64 \times \frac{H}{2} \times \frac{W}{2}$ | $64 \times 8 \times \frac{H}{2} \times \frac{W}{2}$ |
| Spectral Folding | – | – | Depth-wise 3D Conv
$8 \times 1 \times 1$
in_ch=64, groups=64 |
| Post-Folding Output | – | – | $64 \times 1 \times \frac{H}{2} \times \frac{W}{2}$ |
| MaxPooling | ✓ | ✓ | ✓ |
| Final Output | $64 \times \frac{H}{4} \times \frac{W}{4}$ | $64 \times \frac{H}{4} \times \frac{W}{4}$ | $64 \times \frac{H}{4} \times \frac{W}{4}$ |
| Param Count | 9,408
$= 3 \times 64 \times 7 \times 7$ | 25,088
$= 8 \times 64 \times 7 \times 7$ | 9,920
$= 3 \times 64 \times 7 \times 7$
$+ 8 \times 64$ (fold) |

oriented objects. The same detector outputs are used as input to each tracker, isolating the effect of tracking logic and ensuring consistent comparison across paradigms and modalities.

During inference, given the difficulty in detecting small and dense targets, we uniformly set a low detection confidence threshold of 0.1 across all trackers, enabling more candidate boxes to participate in tracking association. While potentially introducing false positives, this threshold maintains fairness and comparability among evaluated methods. All remaining association parameters and inference settings adhere strictly to their original algorithms to preserve consistency.

**Tracking-by-Query Methods.** MOTR and MeMOTR are trained in a single-stage manner. MOTRv2 incorporates pseudo labels from YOLOv11 detectors trained on the training set as detection proposals—using RGB-YOLO for the RGB domain and MSI-YOLO for the MSI domain. MOTIP adopts a two-stage training strategy, where the first stage trains a Deformable-DETR detector independently, and the second stage jointly optimizes detection and ID association.

All models are adapted for rotated bounding boxes using the orientation-aware framework described in this paper. For MSI training, the original ResNet-50 stem in all models is replaced by the proposed Spectral 3D-Stem module (with kernel size $k=7$ and output dimension $D=64$).

Training schedules are as follows: MOTR is trained for 20 epochs; MOTRv2 for 20 epochs (increased from the original 5 epochs); MeMOTR for 20 epochs; and MOTIP for 60 epochs in stage one and 14 epochs in stage two. Learning rates are kept identical to the original settings in the RGB domain. In the MSI domain, the learning rate of the Spectral 3D-Stem is scaled by a factor of 10. For MOTRv2, the learning rate drop is scheduled at epoch 10 to match the increased training length.

Considering the relatively low frame rate of MMOT, the temporal sampling interval is uniformly set to 3 across all query-based trackers. To accommodate the dataset's high target density, we set `NUM_ID_VOCABULARY` to 300 and reduce `SAMPLE_LENGTHS` to 10 for MOTIP.

## A.9 Computational Resources

All experiments were conducted on machines equipped with NVIDIA RTX 3090 GPUs. For YOLOv11, Deformable-DETR-based detectors, and query-based trackers such as MOTR, MOTRv2, MeMOTR, and MOTIP, we employed 2 GPUs for training.

## A.10 Detailed Comparison across Modalities and Stem Variants

To provide a detailed understanding of tracking performance across different input modalities and stem configurations, Table 9 reports superclass-wise metrics for a wide set of MOT algorithms.

The results are reported per superclass (HUMAN, VEHICLE, and BICYCLE), covering five key metrics (HOTA, MOTA, IDF1, DetA, AssA). Each row corresponds to a method from either the tracking-by-detection or tracking-by-query paradigm, fully adapted to handle oriented bounding boxes.

The table reveals that MSI input consistently improves performance across superclasses compared to RGB, especially in human and bicycle categories where spectral cues are more informative. Moreover, the use of Spectral 3D-Stem leads to substantial gains over the 2D-Stem baseline, validating its design for spectral-spatial feature extraction.

To ensure statistical robustness and meet reproducibility standards, we repeated each experiment three times under identical settings and report the mean HOTA score along with its standard deviation (denoted as ±). These results, summarized in the final column ("Cls. Avg."), provide error estimates for key comparisons that support the central claims of this paper.

## A.11 Impact of Detector Quality on Tracking Performance

To investigate how the choice of detector affects the performance of tracking-by-detection (TBD) methods, we compare four representative algorithms—SORT, ByteTrack, OC-SORT, and BoT-SORT—under two detectors: YOLOv11-L and Deformable-DETR (D-DETR). The results are summarized in Table 10.

We observe that YOLOv11-L, with a stronger detection baseline ($mAP_{50} = 73.4$), consistently yields superior tracking performance across all TBD models compared to D-DETR ($mAP_{50} = 62.1$). For instance, under YOLOv11, BoT-SORT achieves a class-averaged HOTA of 53.6 and detection-averaged HOTA of 60.7, while the same model under D-DETR drops to 39.2 and 50.1, respectively. This trend persists across all trackers and evaluation modes, suggesting that detection quality remains a critical bottleneck in tracking performance.

Overall, these results emphasize the strong coupling between detector quality and TBD performance. We further hypothesize that one of the main reasons why tracking-by-query (TBQ) methods currently underperform compared to TBD models on MMOT is due to the relatively limited detection capacity of the end-to-end query-based architectures. Addressing this limitation by integrating more advanced detection modules could be a promising direction for improving TBQ frameworks in future work.

## A.12 Broader Societal Impacts

This work introduces the MMOT dataset and a rotation-aware multispectral tracking framework to advance research in drone-based multi-object tracking. The proposed contributions have several potential positive societal impacts. Enhanced aerial tracking performance can benefit public safety and emergency response operations, such as search and rescue, disaster monitoring, and traffic management, particularly in complex environments where conventional RGB-based systems fail.

However, we also acknowledge potential negative impacts. As with all tracking technologies, misuse for mass surveillance or privacy invasion is a concern. The ability to robustly detect and track small and densely distributed objects raises ethical questions when deployed without adequate oversight. Moreover, the collection of aerial imagery may raise regulatory and societal concerns regarding data consent and usage rights.

To mitigate such risks, we recommend that any deployment of this technology comply with existing legal frameworks and ethical standards for responsible UAV use. We also encourage future work to explore privacy-preserving tracking mechanisms and fair evaluation under diverse demographic and geographic conditions.

Table 9: Per-superclass tracking performance comparison across different input domains and stem designs (RGB, MSI, MSI-2DStem) on the MMOT dataset. Metrics are reported for HUMAN, VEHICLE, and BICYCLE categories. Each method's class-averaged HOTA is reported with standard deviation (±) computed over three independent runs to reflect statistical variability.

| Domain | Method | HUMAN | | | | | VEHICLE | | | | | BICYCLE | | | | | HOTA |
|---|---|---|---|---|---|---|---|---|---|---|---|---|---|---|---|---|---|
| | | HOTA | MOTA | IDF1 | DetA | AssA | HOTA | MOTA | IDF1 | DetA | AssA | HOTA | MOTA | IDF1 | DetA | AssA | Cls. Avg. |
| MSI | SORT [18] | 4.8 | 1.6 | 4.5 | 4.0 | 6.2 | 51.1 | 54.4 | 55.7 | 54.1 | 48.3 | 12.2 | 6.7 | 11.6 | 8.8 | 17.4 | 27.2±0.3 |
| | ByteTrack [19] | 13.3 | 9.4 | 13.5 | 19.4 | 9.3 | 65.5 | 71.0 | 72.4 | 68.0 | 63.3 | 25.6 | 18.9 | 26.4 | 22.1 | 30.3 | 40.5±0.2 |
| | OC-SORT [20] | 5.8 | 2.3 | 5.5 | 4.9 | 7.1 | 54.5 | 57.7 | 60.1 | 57.0 | 52.4 | 13.3 | 7.4 | 13.2 | 10.3 | 17.6 | 29.5±0.2 |
| | BoT-SORT [21] | 43.1 | 48.1 | 54.7 | 45.2 | 41.5 | 76.7 | 77.0 | 86.5 | 71.2 | 82.9 | 43.1 | 31.3 | 48.1 | 35.2 | 53.2 | 53.6±0.3 |
| | MOTR [22] | 26.4 | 0.1 | 29.9 | 19.5 | 36.6 | 66.1 | 67.6 | 78.8 | 57.7 | 76.5 | 32.3 | 20.3 | 37.1 | 18.2 | 58.0 | 39.0±0.4 |
| | MOTRv2 [23] | 36.4 | 34.2 | 49.6 | 30.0 | 44.5 | 70.1 | 72.7 | 80.6 | 64.5 | 76.5 | 39.0 | 29.3 | 45.4 | 24.4 | 63.0 | 49.2±0.6 |
| | MeMOTR [24] | 31.6 | 25.1 | 38.0 | 22.5 | 44.6 | 66.8 | 61.0 | 73.8 | 56.7 | 78.9 | 35.6 | 21.7 | 38.4 | 19.4 | 65.5 | 42.3±0.3 |
| | MOTIP [25] | 26.3 | 19.5 | 29.9 | 28.9 | 24.4 | 56.5 | 59.2 | 61.3 | 64.0 | 50.4 | 32.7 | 20.5 | 38.4 | 24.3 | 44.5 | 39.0±0.4 |
| RGB | MOTR [22] | 19.4 | -0.7 | 19.2 | 12.4 | 31.6 | 64.6 | 66.2 | 76.4 | 57.0 | 73.9 | 30.4 | 17.9 | 34.9 | 17.0 | 55.3 | 39.3±0.4 |
| | MOTRv2 [23] | 29.0 | 23.3 | 37.9 | 21.5 | 39.7 | 67.1 | 70.4 | 77.1 | 62.9 | 72.0 | 37.2 | 26.8 | 42.5 | 22.6 | 61.8 | 45.9±0.4 |
| | MeMOTR [24] | 24.3 | 17.4 | 26.9 | 15.5 | 38.3 | 63.5 | 58.2 | 70.7 | 54.3 | 74.5 | 34.3 | 20.8 | 36.5 | 18.3 | 64.6 | 41.6±0.3 |
| MSI-2D Stem | ByteTrack [19] | 12.9 | 10.0 | 13.4 | 19.2 | 9.0 | 66.5 | 74.2 | 73.3 | 70.1 | 63.3 | 24.7 | 18.2 | 25.6 | 20.9 | 29.9 | 40.3±0.2 |
| | BoT-SORT [21] | 42.3 | 47.7 | 53.2 | 45.0 | 40.3 | 78.7 | 81.7 | 88.7 | 74.3 | 83.7 | 41.2 | 29.3 | 45.5 | 33.6 | 50.8 | 52.8±0.3 |
| | MOTR [22] | 20.6 | 0.0 | 20.4 | 12.8 | 34.3 | 59.7 | 56.9 | 70.3 | 49.5 | 72.5 | 27.0 | 10.8 | 28.4 | 12.5 | 59.1 | 35.9±0.5 |
| | MeMOTR [24] | 29.8 | 22.6 | 34.5 | 20.3 | 44.1 | 68.1 | 63.8 | 75.9 | 58.6 | 79.4 | 35.8 | 21.1 | 38.5 | 19.0 | 67.8 | 38.5±0.3 |

Table 10: Comparison of tracking-by-detection methods under two different detectors (YOLOv11-L and Deformable-DETR) on the MMOT dataset. Metrics are reported for class-averaged and detection-averaged settings.

| Detector | Tracker | Class-Averaged | | | | | Detection-Averaged | | | | |
|---|---|---|---|---|---|---|---|---|---|---|---|
| | | HOTA | MOTA | IDF1 | DetA | AssA | HOTA | MOTA | IDF1 | DetA | AssA |
| MSI YOLOv11 [26] $mAP_{50} = 73.4$ | SORT [18] | 27.2 | 24.3 | 29.1 | 25.7 | 30.0 | 27.2 | 24.3 | 29.1 | 25.7 | 30.0 |
| | ByteTrack [19] | 40.5 | 34.2 | 44.1 | 37.0 | 46.2 | 46.0 | 37.8 | 46.7 | 41.9 | 51.5 |
| | OC-SORT [20] | 29.5 | 25.1 | 31.9 | 27.3 | 32.8 | 37.5 | 27.5 | 37.0 | 29.5 | 48.0 |
| | BoT-SORT [21] | 53.6 | 46.2 | 61.0 | 45.7 | 64.6 | 60.7 | 59.4 | 69.4 | 55.0 | 68.7 |
| MSI D-DETR [28] $mAP_{50} = 62.1$ | SORT [18] | 21.2 | 18.0 | 23.1 | 19.0 | 25.9 | 29.4 | 21.3 | 28.8 | 22.5 | 38.8 |
| | ByteTrack [19] | 33.0 | 25.8 | 36.7 | 29.0 | 40.5 | 40.8 | 29.7 | 41.9 | 35.2 | 48.5 |
| | OC-SORT [20] | 23.4 | 19.2 | 26.2 | 20.8 | 28.5 | 31.7 | 23.0 | 31.6 | 24.4 | 41.7 |
| | BoT-SORT [21] | 39.2 | 31.0 | 44.7 | 31.6 | 51.1 | 50.1 | 39.7 | 55.8 | 38.6 | 66.2 |

## A.13 Licenses for Existing Assets

Our work builds upon several open-source software implementations and public datasets. We summarize below all models, datasets and software along with their corresponding licenses in Tab. 11.

Table 11: Summary of external software and dataset assets reused in our work. All resources are used under their original licenses and for academic research only.

| Asset | URL | Usage in Our Work |
|---|---|---|
| YOLOv11 [26] | https://github.com/ultralytics/ultralytics | Detector for TBD methods |
| SORT [18] | https://github.com/abewley/sort | Tracking-by-detection baseline |
| ByteTrack [19] | https://github.com/ifzhang/ByteTrack | Tracking-by-detection baseline |
| OC-SORT [20] | https://github.com/noahcao/OC_SORT | Tracking-by-detection baseline |
| BoT-SORT [21] | https://github.com/yezzed/BoT-SORT | Tracking-by-detection baseline |
| MOTR [22] | https://github.com/megvii-research/MOTR | Tracking-by-query baseline |
| MOTRv2 [23] | https://github.com/megvii-research/MOTRv2 | Tracking-by-query baseline |
| MeMOTR [24] | https://github.com/MCG-NJU/MeMOTR | Tracking-by-query baseline |
| MOTIP [25] | https://github.com/MCG-NJU/MOTIP | Tracking-by-query baseline |
| TrackEval [33] | https://github.com/JonathonLuiten/TrackEval | Evaluation framework |
| UAVDT [4] Dataset | https://sites.google.com/view/grli-uavdt/ | Statistical comparison |
| VisDrone [5] Dataset | https://github.com/VisDrone/VisDrone-Dataset | Statistical comparison |
| X-AnyLabeling [32] | https://github.com/CVHub520/X-AnyLabeling | Re-development for OBB-MOT annotation |

