# OpenReview forum: "MMOT: The First Challenging Benchmark for Drone-based Multispectral Multi-Object Tracking"
_NeurIPS.cc/2025/Datasets_and_Benchmarks_Track — NeurIPS 2025 Datasets and Benchmarks Track poster_

### Official Review · Reviewer_QRJ8 · 2025-06-29

**Rating:** 5
**Confidence:** 5

**Summary:**

This paper introduces MMOT, a novel large-scale benchmark for drone-based multi-object tracking (MOT) using multispectral imagery. The primary contribution is the dataset itself, which is the first of its kind, featuring sequences, annotations with precisely oriented bounding boxes (OBBs) across eight object categories. The dataset is specifically designed to address challenging real-world scenarios such as small targets, severe occlusions, and high-density scenes, where traditional RGB-based methods often fail. In addition to the dataset, the authors propose a unified adaptation scheme, including a Spectral 3D-Stem module, to enable existing MOT models to effectively leverage multispectral and orientation information. A comprehensive benchmark of eight representative MOT trackers is provided, demonstrating significant performance gains from multispectral input, particularly for small and densely packed objects.

**Additional Feedback:**

No.

**Dataset Code Accessibility:**

Yes

**Dataset Code Comments:**

No.

**Ethical Considerations:**

No, there are no or only very minor ethics concerns

**Final Justification:**

The authors solved all of my concerns.
I am willing to rate this paper 5.

The authors should consider adding some of my questions and their answers to the supplementary materials in order to improve the representation of this paper.

Thanks all authors, AC, PC.

**Limitations Weaknesses:**

1. Long-tail Distribution: The paper's statistical analysis (Fig. 4e, 4f) correctly identifies a significant long-tail problem in both class distribution and trajectory length. This is a well-known challenge that often requires specialized techniques like re-sampling, re-weighting, or dedicated long-tail learning strategies. However, the proposed Spectral 3D-Stem and orientation-aware adaptations do not directly address this long-tail issue.
2. Lack of Explicit Occlusion Handling: Severe occlusion is listed as a key challenge in the abstract and introduction. Yet, the proposed methodological adaptations—focused on feature extraction and state estimation—do not include any explicit mechanisms for handling occlusions, such as trajectory interpolation, re-identification modules robust to partial views, or memory-augmented networks (like the benchmarked MeMOTR, whose strengths are not fully analyzed in this context). The paper benchmarks models but does not offer novel insights on how to solve the occlusion problem with multispectral data.
3. Kalman Filter Extension: The extension of the Kalman filter state vector to include orientation and its velocity (Sec. 4.2) is a straightforward adaptation. The paper provides no analysis or experiments to demonstrate how much this explicit orientation modeling actually improves tracking performance (e.g., in terms of reducing ID switches or improving localization) compared to a baseline that uses OBB for association but a standard Kalman filter for motion prediction.
4.  Lack of Alternative Fusion Baselines: A more rigorous comparison would involve testing against other simple but common multispectral fusion techniques. For instance, a baseline could involve channel-wise averaging or a principal component analysis (PCA) to reduce the 8 channels to 3 before feeding them into a standard RGB model. By only comparing against a single, arbitrarily chosen pseudo-RGB combination, the paper doesn't fully demonstrate that its proposed Spectral 3D-Stem is superior to other, simpler fusion strategies.
5. Band Selection Justification: The paper states that bands 5, 3, and 2 are selected because they "approximately correspond to the RGB spectrum." This claim is not substantiated with any spectral response analysis or reference. Different band combinations could yield vastly different "RGB" appearances and performance. The chosen combination might inadvertently be a weak one, thus exaggerating the performance gap with the 8-band MSI input.

**Strengths Contributions:**

The key strengths and contributions of this work are significant and highly relevant to the computer vision remote sensing area:
1.  To the best of my knowledge, MMOT is the first large-scale, publicly available dataset for drone-based MOT that combines multispectral imaging with precise oriented bounding box (OBB) annotations.
2. It provides a solid foundation for training and evaluating deep learning models. The choice of OBB annotations is particularly well-suited for aerial perspectives, as it more accurately represents object geometry and reduces ambiguity compared to standard axis-aligned boxes.
3. The authors do not just release a dataset; they provide a comprehensive benchmark by adapting and evaluating eight state-of-the-art MOT algorithms. This sets a strong baseline for future research.

---

> ### Author Rebuttal · Authors · 2025-07-31
>
> # Response to reviewer QRJ8's comments
>
> Thank you for your recognition of our work. Your comments have helped us significantly improve the clarity and depth of our work. Our responses below aim to address your concern.
>
> > **Q1: Long-tail Distribution: The paper's statistical analysis (Fig. 4e, 4f) correctly identifies a significant long-tail problem in both class distribution and trajectory length. This is a well-known challenge that often requires specialized techniques like re-sampling, re-weighting, or dedicated long-tail learning strategies. However, the proposed Spectral 3D-Stem and orientation-aware adaptations do not directly address this long-tail issue.**
>
> Thank you for your suggestion. Our dataset is designed to reflect the challenges of real-world urban scenarios, where long-tail distributions in both class frequencies and trajectory lengths natural occur, as also observed in existing benchmarks such as VisDrone-MOT.
>
> We acknowledge that addressing the long-tail problem is of great importance. Our core motivation is to establish the benchmark for multispectral multi-object tracking (MOT) with oriented bounding boxes, thereby providing a realistic benchmark and a group of baselines for the community. We believe that such a benchmark with long-tail distribution will enable the development of methods that can effectively handle the challenges of real-world. We also plan to further investigate dedicated strategies for long-tail learning in future research.
>
>
>
> > **Q2: Lack of Explicit Occlusion Handling: Severe occlusion is listed as a key challenge in the abstract and introduction. Yet, the proposed methodological adaptations—focused on feature extraction and state estimation—do not include any explicit mechanisms for handling occlusions, such as trajectory interpolation, re-identification modules robust to partial views, or memory-augmented networks (like the benchmarked MeMOTR, whose strengths are not fully analyzed in this context). The paper benchmarks models but does not offer novel insights on how to solve the occlusion problem with multispectral data.**
>
> We sincerely thank you for this valuable suggestion. When severe occlusion occurs, the spatial features of the target become severely constrained. In such cases, multispectral imagery provides pixel-wise spectral measurements, which can serve as complementary cues that remain distinguishable from the background. This enables the model to better maintain target separability, even under heavy occlusion.
>
> To validate this hypothesis, we filtered the severely occluded targets from the MMOT test set to construct a subset, which could reflect the models' capability to handle occlusion. On this subset, we compared the percentage of successfully matched ground-truth instances (TP/GT) under different input modalities.  As shown in the table below, multispectral inputs increase true positive percentage by **3.8%**, demonstrating that spectral information enables more accurate target localization. As a result, spectral information serves as a powerful cue for addressing occlusion, effectively compensating for the spatial feature degradation caused by severe overlaps.
>
> We agree that explicitly leveraging spatial–spectral information to address occlusion is a promising direction. Our dataset and baselines establish a solid benchmark for the community to further develop multispectral tracking algorithms for occlusion handling, and we also plan to actively explore this line of research in our future work.
>
> | Model  | Input |   TP/GT(↑)   |
> | :----: | :---: | :----------: |
> | MeMOTR |  RGB  |    41.5%     |
> |        |  MSI  | 45.3%(↑3.8%) |
>
>
> > **Q3: Kalman Filter Extension: The extension of the Kalman filter state vector to include orientation and its velocity (Sec. 4.2) is a straightforward adaptation. The paper provides no analysis or experiments to demonstrate how much this explicit orientation modeling actually improves tracking performance (e.g., in terms of reducing ID switches or improving localization) compared to a baseline that uses OBB for association but a standard Kalman filter for motion prediction.**
>
> Thank you for raising this point. Following your suggestion, we compared the performance of the **standard Kalman filter**, the Kalman filter with **orientation parameter**, and the Kalman filter with **both orientation and angular velocity parameters**. As shown in the table below, explicitly modeling orientation and angular velocity yields consistent and substantial improvements. In aerial scenes, where targets can appear at arbitrary orientations, this explicitly modeling enables a more precise representation of target states, thereby improving tracking performance.
>
> | Model     | Including Orientation Parameter | Including Angular Velocity Parameter | HOTA(↑)  | MOTA(↑)  | IDF1(↑)  |
> | --------- | :-----------------------------: | :----------------------------------: | :------: | :------: | :------: |
> | ByteTrack |                ❌                |                  ❌                   |   35.6   |   30.3   |   37.7   |
> |           |                ✔️                |                  ❌                   |   40.0   |   34.2   |   43.5   |
> |           |                ✔️                |                  ✔️                   | **40.5** | **34.2** | **44.1** |
> | Bot-SORT  |                ❌                |                  ❌                   |   51.5   |   42.6   |   56.9   |
> |           |                ✔️                |                  ❌                   |   52.7   |   45.1   |   59.3   |
> |           |                ✔️                |                  ✔️                   | **53.6** | **46.2** | **61.0** |
>
>
>
> > **Q4: Lack of Alternative Fusion Baselines: A more rigorous comparison would involve testing against other simple but common multispectral fusion techniques. For instance, a baseline could involve channel-wise averaging or a principal component analysis (PCA) to reduce the 8 channels to 3 before feeding them into a standard RGB model. By only comparing against a single, arbitrarily chosen pseudo-RGB combination, the paper doesn't fully demonstrate that its proposed Spectral 3D-Stem is superior to other, simpler fusion strategies.**
>
> Thank you for this valuable suggestion. To provide a comprehensive evaluation, we implemented PCA and channel-wise averaging as alternative baselines alongside our pseudo RGB combination. Among all evaluated methods, **the proposed Spectral 3D-Stem achieves the best results**. We sincerely appreciate your suggestion, as these additional experiments further confirm the effectiveness of our proposed Spectral 3D-Stem module in leveraging spectral information for robust multispectral tracking
>
> |  Method   | Multispectral Fusion Method |   HOTA   |   MOTA   |   IDF1   |
> | :-------: | :-------------------------- | :------: | :------: | :------: |
> | ByteTrack | PCA                         |   39.2   |   32.8   |   42.7   |
> |           | Channel-wise averaging      |   38.6   |   33.1   |   41.0   |
> |           | Pseudo RGB                  |   40.0   |   33.5   |   43.4   |
> |           | **Spectral 3D-Stem**        | **40.5** | **34.2** | **44.1** |
> | Bot-SORT  | PCA                         |   50.7   |   42.5   |   56.9   |
> |           | Channel-wise averaging      |   50.2   |   41.8   |   55.3   |
> |           | Pseudo RGB                  |   52.6   |   44.3   |   59.1   |
> |           | **Spectral 3D-Stem**        | **53.6** | **46.2** | **61.0** |
>
>
>
> > **Q5: Band Selection Justification: The paper states that bands 5, 3, and 2 are selected because they "approximately correspond to the RGB spectrum." This claim is not substantiated with any spectral response analysis or reference. Different band combinations could yield vastly different "RGB" appearances and performance. The chosen combination might inadvertently be a weak one, thus exaggerating the performance gap with the 8-band MSI input.**
>
> We appreciate your suggestion for clarification. Below, we present the detailed spectral wavelength of our multispectral images. We chose bands **5 (660 nm)**, **3 (550 nm)**, and **2 (487.5 nm)** explicitly because their center wavelengths closely match the canonical RGB centers **(red 685 nm, green 532.5 nm, blue 472.5 nm)**. This alignment ensures that our selected 5-3-2 combination is both physically meaningful and well-suited to RGB-based models.
>
> Furthermore, we trained and evaluated our model using several alternative band combinations. Among all tested combinations, **bands 5-3-2 achieved the best results**, validating our choice from both a physical and an experimental perspective.
>
> | Bands | Start(nm) | End(nm) | Center(nm) |  Color   |
> | :---: | :-------: | :-----: | :--------: | :------: |
> |   1   |    395    |   450   |   422.5    |  Violet  |
> |   2   |    455    |   520   |   487.5    |   Blue   |
> |   3   |    525    |   575   |   550.0    |  Green   |
> |   4   |    580    |   625   |   602.5    |  Orange  |
> |   5   |    630    |   690   |   660.0    |   Red    |
> |   6   |    705    |   745   |   725.0    | Red Edge |
> |   7   |    750    |   820   |   785.0    |   NIR1   |
> |   8   |    825    |   950   |   887.2    |   NIR2   |
>
> | Band Combination | HTOA of Bot-SORT |
> | :--------------: | :--------------: |
> |      4,2,1       |       52.0       |
> |      6,3,1       |       51.7       |
> |      6,4,3       |       52.4       |
> |      7,5,2       |       28.5       |
> |   5,3,2 (ours)   |     **52.6**     |
>
>
>
> We sincerely thank you again for your constructive feedback and hope that our detailed responses adequately addressed your concern.

---

> > ### Comment · Reviewer_QRJ8 · 2025-08-03
> >
> > Thanks for the authors' reply. The rebuttal solved my concerns to some extent.
> >
> > I still have one question about Q2:
> > It seems that involving some memory modules may improve the performance. Can the authors comment on this aspect?
> > The table for Q2 shows that the improvement only comes from the MSI property. I wonder if the tracking boxes are annotated on RGB images (I'm not sure about this) by human experts, will it limit the model's performance? Please correct me if I'm wrong.
> >
> > I keep the rating to 4 (borderline accept) right now.

---

> > > ### Author Response · Authors · 2025-08-04
> > >
> > > We sincerely appreciate your detailed reviews. We will address your questions point by point.
> > >
> > > > **Discussion Q1: It seems that involving some memory modules may improve the performance.**
> > >
> > > Based on MOTR, **MeMOTR** [1] introduces an end-to-end memory-augmented MOT framework. Compared with MOTR, it incorporates three key innovations:
> > >
> > > (1) **Long-term memory**, which explicitly maintains a persistent memory for each target across frames;
> > >
> > > (2) **A temporal-interaction module**, which adaptively fuses outputs from two adjacent frames using an adaptive aggregation algorithm;
> > >
> > > (3) **A memory-attention layer**, which establishes interactions between different trajectories and generates updated track embeddings.
> > >
> > > The long-term memory significantly stabilizes the identity information of each target, thereby improving the consistency of track embeddings. Meanwhile, the temporal-interaction module provides complementary feature augmentation that mitigates video ambiguity and uncertainty. Finally, the memory-attention layer enables cross-trajectory interactions, helping track embeddings learn more discriminative features.
> > >
> > > Through these three innovations, MeMOTR establishes reliable and accurate target association, which is crucial for multi-object tracking. As a result, it achieves consistent improvements over its MOTR baseline, as shown in Table 3 of our paper, which is also consistent with the results reported in the original MeMOTR paper on other datasets.
> > >
> > >
> > >
> > > > **Discussion Q2: The table for Q2 shows that the improvement only comes from the MSI property. I wonder if the tracking boxes are annotated on RGB images (I'm not sure about this) by human experts, will it limit the model's performance?**
> > >
> > > Thank you for raising this point. We would like to clarify our annotation process.
> > >
> > > Our MSI images consist of 8 spectral channels, which, as described in Section 3.3 of our paper, are precisely co-registered to ensure **pixel-level spatial alignment across all channels**. The dataset was manually annotated for accuracy, and to better match human visual perception, the tracking boxes were annotated on pseudo-RGB images composed of bands 5-3-2. Owing to the precise co-registration, these annotations can be directly propagated to all multispectral channels without spatial misalignment, **ensuring their suitability for training and evaluating both RGB-based and MSI-based models**. **Therefore, this annotation strategy does not limit the performance of either MSI-based or RGB-based models.**
> > >
> > >
> > >
> > > **References:**
> > >
> > > [1] Gao R, et al. MeMOTR: Long-term memory-augmented transformer for multi-object tracking[C]//Proceedings of the IEEE/CVF International Conference on Computer Vision. 2023.

---

> > > > ### Comment · Reviewer_QRJ8 · 2025-08-04
> > > >
> > > > Thanks for the quick reply.
> > > >
> > > > I mean, for instance, if certain objects can only be detected in specific wavelength bands—other than the standard 5-3-2 pseudo-true-color RGB combination—relying solely on the 5-3-2 bands for human annotation may result in those objects being overlooked or appearing unconvincing in their trajectories.
> > > >
> > > > Maybe this scene is rare in the real world?

---

> > > > > ### Author Response · Authors · 2025-08-04
> > > > >
> > > > > Thanks for your detailed reviews.
> > > > >
> > > > > In real-world scenarios, there are indeed cases where targets exhibit uneven discriminability across different spectral channels. To address potential ambiguities that may arise if annotations relied solely on the 5-3-2 pseudo–true-color images, we have implemented the following measures to ensure annotation accuracy and spatiotemporal continuity:
> > > > >
> > > > > 1. When a target is not sufficiently discernible in the 5-3-2 composite image, annotators examine other spectral channels to identify the channel in which the target is most distinguishable and use it to determine the target’s existence, spatial position, and boundaries.
> > > > > 2. When the target’s boundaries remain unclear, annotators apply localized zooming as well as brightness and contrast adjustments to improve boundary visibility and assist in precise annotation.
> > > > > 3. When a target cannot be confidently identified from a single frame, annotators review the entire video sequence and leverage the target’s temporal context to determine its existence and spatial position, thereby ensuring spatiotemporal continuity in the annotations.
> > > > >
> > > > > Additionally, during the ID annotation stage, each trajectory is carefully reviewed to ensure temporal continuity and semantic correctness. Annotators are required to clearly describe the reasons for any target’s appearance or disappearance, such as complete occlusion or entering or leaving the field of view.
> > > > >
> > > > > Through these measures, our annotation process incorporates both multi-spectral information and temporal context, thereby ensuring high precision, unambiguity, and spatiotemporal continuity in the dataset.
> > > > >
> > > > > We hope that our detailed responses addressed your concern.

---

> > > > > > ### Comment · Reviewer_QRJ8 · 2025-08-04
> > > > > >
> > > > > > Thanks for the authors' detailed response. I have no more questions. I will raise the rating to 5.

---

> > > > > > > ### Author Response · Authors · 2025-08-04
> > > > > > >
> > > > > > > Thanks again for your great efforts in reviewing this paper and for your recognition of our work.

---

### Official Review · Reviewer_47W5 · 2025-07-03

**Rating:** 4
**Confidence:** 3

**Summary:**

This paper presents MMOT, the first large-scale benchmark for drone-based multispectral multi-object tracking. The primary contributions are threefold: First, the construction of a multispectral dataset with 125 video sequences and 488K annotations across 8 spectral bands. Second, a unified adaptation scheme that enables existing MOT algorithms to handle multispectral inputs and oriented bounding boxes, including a Spectral 3D-Stem module, orientation-aware Kalman filter, and end-to-end orientation-adaptive transformer. Third, comprehensive experimental evaluation on 8 representative MOT algorithms demonstrating consistent performance improvements of multispectral input over RGB baselines.

**Dataset Code Accessibility:**

Yes

**Ethical Considerations:**

No, there are no or only very minor ethics concerns

**Final Justification:**

After reviewing the authors' rebuttal, I have decided to raise my score from 3 (Borderline reject) to 4 (Borderline accept).

The authors successfully addressed my two main concerns with strong experimental evidence. They demonstrated that multispectral information provides a 150% improvement in inter-ID discriminability and ~3-point gains in association accuracy, convincingly showing enhanced target separability. Additionally, their DOTA pretraining experiments proved that RGB-trained aerial models can effectively transfer to multispectral data, addressing the limited training data issue. While the physical property modeling could be more sophisticated, this is acceptable as future work. The paper's contribution of the first large-scale multispectral MOT benchmark, now backed by stronger technical validation, merits publication.

**Limitations Weaknesses:**

1. Insufficient Demonstration of Multispectral Advantages: As pointed out by Reviewer QXcp, while Figure 1(b) shows that spectral information can separate targets from background, I agree that the discriminability between targets remains limited. The paper lacks deep analysis of the physical properties of multispectral images or principles of spectral reflectance. There is insufficient clear evidence whether it can satisfy the performance requirements for low-resolution, high-density targets at long distances.

2. Utilization of RGB-trained Neural Networks: RGB images are also capable of extracting features from vehicles or objects. Given the limited availability of spectral image data, it would be valuable to experimentally demonstrate whether existing RGB-trained neural networks can be effectively adapted to spectral image networks.

3. Insufficient Physical Property Modeling: The proposed Spectral 3D-Stem module does not adequately reflect the physical properties of multispectral images. It fails to explicitly model inter-band correlations, material-specific spectral characteristics, or wavelength-specific physical meanings, instead simply processing through Conv3D operations. This represents a limitation in maximizing the inherent advantages of multispectral imagery. However, this could be proposed as future research.

Additional experiments addressing points 1 and 2 would strengthen the paper significantly. If the authors provide appropriate responses to these two issues, I would be willing to offer acceptance.

**Strengths Contributions:**

1. Dataset Innovation and Quality: The paper's most significant strength lies in constructing the first dedicated dataset for drone-based multispectral MOT. The dataset ensures high quality through manual annotation and represents the only multispectral MOT dataset providing oriented bounding box annotations. With 8 spectral bands and diverse challenging scenarios, it provides realistic data suitable for practical applications.

2. Comprehensive Experimental Evaluation: The systematic evaluation across 8 different MOT algorithms consistently demonstrates the effectiveness of multispectral input. The significant performance improvement particularly for small and dense objects proves its utility in real-world applications.

3. Paper Organization and Clarity: The paper is systematically organized with clear presentation of dataset construction, technical solutions, and experimental results. Figure 1 and Table 1 effectively demonstrate the advantages of multispectral images and distinctions from existing datasets.

---

> ### Author Rebuttal · Authors · 2025-07-31
>
> # Response to reviewer 47W5's comments
>
> We sincerely appreciate your detailed and constructive feedback. We carefully address your concerns below and hope our response resolves your questions. We look forward to discussing any issues further should you have any follow-up concerns!
>
>
>
> > **Q1: Insufficient Demonstration of Multispectral Advantages: As pointed out by Reviewer QXcp, while Figure 1(b) shows that spectral information can separate targets from background, I agree that the discriminability between targets remains limited. The paper lacks deep analysis of the physical properties of multispectral images or principles of spectral reflectance. There is insufficient clear evidence whether it can satisfy the performance requirements for low-resolution, high-density targets at long distances.**
>
> Thank you for your suggestion. Multispectral imagery not only captures spatial features but also provides **pixel-wise spectral measurements**. These spectral cues act as a complementary source of information to spatial features, enhancing the discriminability between different targets. And these pixel-wise spectral responses become particularly valuable in **low-resolution and high-density tracking scenarios**, where appearance cues alone are insufficient for distinguishing between visually similar targets.
>
> To verify this hypothesis, we conducted the following experiments to quantitatively assess the impact of spectral information on target discriminability in UAV-based multi-object tracking.
>
> **1. Association Metrics Improvement Using Multispectral Inputs**
>
> Target discriminability is crucial for ensuring that a tracking model can correctly associate targets across frames. The higher the discriminability between different targets, the better the model's association performance. Therefore, we evaluated MOTRv2 and MeMOTR under both RGB and multispectral (MSI) inputs, using HOTA and its two sub-metrics: DetA and AssA as metrics. DetA measures detection accuracy, while **AssA evaluates how accurately predicted trajectories align with ground-truth trajectories and how consistently they maintain target identities across frames**.
>
> As shown in the table, both MOTRv2 and MeMOTR exhibit consistent improvements when using multispectral inputs. Notably, besides HOTA and DetA, **the AssA metric improves by approximately 3.0 points**. These results suggest that spectral information particularly enhances discriminability between different targets.
>
> | Model  | Input | HOTA(↑) | DetA(↑) |    AssA(↑)     |
> | :----: | :---: | :-----: | :-----: | :------------: |
> | MOTRv2 |  RGB  |  50.5   |  39.6   |    **65.6**    |
> |        |  MSI  |  54.5   |  44.1   | **68.8(↑3.2)** |
> | MeMOTR |  RGB  |  47.0   |  32.9   |    **67.9**    |
> |        |  MSI  |  50.9   |  37.1   | **70.9(↑3.0)** |
>
> **2. Inter-ID Discriminability: Local Pairwise Cosine Distance**
>
> In multi-object tracking (MOT), the **main challenge in the association stage lies in the similarity of same-class targets within local regions**. With the introduction of spectral information, the discriminability among these same-class targets can be significantly enhanced.
>
> To validate this hypothesis, in each frame, we computed the cosine distance between all pairs of targets belonging to the same class, considering only those pairs whose spatial distance was smaller than a predefined threshold. Finally, we averaged the cosine distances over all such pairs to obtain the **AvgCosDist** metric. This metric reflects the **degree of separability**: a higher value indicates stronger spectral distinctiveness between targets, whereas a lower value  implies that different targets are more likely to be confused.
>
> Specifically, we report results for a threshold of 200px. As shown in the table, MSI improves the average cosine distance by **150% compared to RGB**, demonstrating that spectral information substantially increases inter-target separability.
>
> |    Metric     | $d_{th}$ |  RGB  |     MSI      |
> | :-----------: | :------: | :---: | :----------: |
> | AvgCosDist(↑) |  200px   | 0.016 | 0.040(↑150%) |
>
> **Conclusion:**
>
> The above experiments—from model-level tracking metrics and inter-ID spectral discriminability—jointly demonstrate that the integration of spectral features improves **target-to-target separability**. Furthermore, as noted by reviewer Qxcp, **Figure 1(b)** in the paper clearly shows that spectral information also enhances **target–background discrimination**. These properties make multispectral information an effective complement to spatial features, thereby offering a robust foundation for addressing real-world challenges.
>
>
>
> > **Q2: Utilization of RGB-trained Neural Networks: RGB images are also capable of extracting features from vehicles or objects. Given the limited availability of spectral image data, it would be valuable to experimentally demonstrate whether existing RGB-trained neural networks can be effectively adapted to spectral image networks.**
>
> Thank you for this excellent suggestion. We fully agree that it is crucial to evaluate the adaptability of RGB-trained models to multispectral data, especially in light of the limited availability of annotated spectral datasets.
>
> - **Leveraging the DOTA Dataset to Enhance Spectral Generalization**
>
> To this end, we leverage the **DOTA[1] dataset**, an aerial-view oriented bounding box detection dataset similar to our task. We first pretrain models on DOTA and then finetune them on our multispectral dataset. As shown in the table below, models initialized with DOTA-pretrained weights consistently outperform those trained from ImageNet in terms of HOTA, MOTA, and IDF1, demonstrating the effectiveness of DOTA pretraining for MSI-based MOT.
>
> |   Model   | Pretrained on DOTA |    HOTA(↑)     |    MOTA(↑)     |    IDF1(↑)     |
> | :-------: | :----------------: | :------------: | :------------: | :------------: |
> | ByteTrack |         ❌          |      40.5      |      34.2      |      44.1      |
> |           |         ✔️          | **41.8(↑1.3)** | **39.2(↑5.0)** | **46.3(↑1.8)** |
> | Bot-SORT  |         ❌          |      53.6      |      46.2      |      61.0      |
> |           |         ✔️          | **55.5(↑1.9)** | **49.4(↑3.2)** | **62.9(↑1.9)** |
> |  MOTRv2   |         ❌          |      49.2      |      43.1      |      57.3      |
> |           |         ✔️          | **51.9(↑2.7)** | **46.1(↑3.0)** | **60.4(↑3.1)** |
>
>
>
> > **Q3: Insufficient Physical Property Modeling: The proposed Spectral 3D-Stem module does not adequately reflect the physical properties of multispectral images. It fails to explicitly model inter-band correlations, material-specific spectral characteristics, or wavelength-specific physical meanings, instead simply processing through Conv3D operations. This represents a limitation in maximizing the inherent advantages of multispectral imagery. However, this could be proposed as future research.**
>
> We appreciate your valuable critique. The proposed **Spectral 3D-Stem** module is composed of two sequential 3D convolutional layers. First, a 3D convolution with a spectral kernel size of 3 and a spatial kernel size of $k\times k$ (where $k$ is identical to the spatial kernel size of the first convolution in the RGB backbone, ie., $7\times 7$ for ResNet-50) is applied. The layer **slides across the spectral axis**, producing 8 groups of feature maps, each derived from three consecutive spectral band. Second, another 3D convolution with a spectral kernel size of 8 and a spatial kernel size of 1$\times$1 is employed. Taking the 8 feature groups from the first layer as input, this second layer aggregates information across the full spectral range thereby producing a spectrally fused feature representation.
>
> Through this design, our Spectral 3D-Stem achieves progressive modeling of spectral relationships, from local to global inter-band correlation. Besides, the first 3D convolution can fully inherit RGB-pretrained weights ensuring efficient initialization.
>
> We fully recognize that incorporating physically grounded spectral modeling, such as wavelength-dependent material properties and spectral priors, is an important research direction. We will investigate it in future work to further enhance the interpretability and physical relevance of our approach.
>
>
>
> We greatly appreciate your insightful comments and hope that our response has addressed your concerns. Should you have any further questions, we would be glad to engage in additional discussion.
>
> **References:**
>
> [1] Xia G, et al. DOTA: A large-scale dataset for object detection in aerial images[C]//Proceedings of the IEEE conference on computer vision and pattern recognition. 2018.

---

> > ### Comment · Area_Chair_4h4T · 2025-08-04
> >
> > Dear Reviewer
> >
> > The Author-Reviewer discussion phase is until 6 Aug. The author has submitted a rebuttal. Please feel free to initialize discussions with the authors.
> >
> > Kind regards
> > AC

---

> > ### Comment · Area_Chair_4h4T · 2025-08-05
> >
> > Dear Reviewers
> >
> > Thanks for contributing to Neurips. We have received new information from the PCs:
> > “Reviewers must participate in discussions with authors before submitting “Mandatory Acknowledgement”. ” “To facilitate discussions, we extend Author-Reviewer discussions by 48h till Aug 8, 11.59pm AoE. ”
> >
> > As informed, please engage in discussions with the authors.
> >
> > Kind regards
> > AC

---

> > ### Comment · Reviewer_47W5 · 2025-08-07
> >
> > Thank you for the comprehensive rebuttal. I am satisfied that my main concerns have been adequately addressed. The additional experiments and clarifications have significantly strengthened the paper.

---

> > > ### Author Response · Authors · 2025-08-07
> > >
> > > We sincerely appreciate your great efforts in reviewing this paper and your recognition of our work.

---

### Official Review · Reviewer_QXcp · 2025-07-04

**Rating:** 5
**Confidence:** 4

**Summary:**

In this paper, a drone-based multi-object tracking based on multispectral images is constructed, aiming to use the physical characteristics of multispectral images to achieve more accurate multi-object tracking under the conditions of low resolution, dense targets and small targets.It is of great significance to promote the multi-object tracking based on UAV.

**Dataset Code Accessibility:**

Yes

**Dataset Code Comments:**

The ReadMe needs to supplement the description of the dataset in the dataset ULR.

**Ethical Considerations:**

No, there are no or only very minor ethics concerns

**Final Justification:**

The authors solve my concerns. After the author's rebuttal and discussions with other reviewers, I still give it a 5 rating (accept).

**Limitations Weaknesses:**

However, the advantages of multispectral images have not been highlighted. Although Figure 1 b shows that the spectrum can separate the target and the background, the discrimination between the targets is still very low. Can it meet the recognition requirements of low-resolution, high-density targets and complex backgrounds for long-distance targets?

Besides, there are logical problems in the section on Inter-frame Motion and Overlap Analysis of Statistical Analysis in Section 3.4. The author points out that the inter-frame IoU of the MMOT dataset is significantly lower than that of other unmanned aerial vehicle datasets (mostly <0.1), and attributes it to the disruption of motion continuity caused by the small target size and high movement speed. However, the explanation for this phenomenon has the following deficiencies:

The distribution consistency of the target's movement speed is questionable: If the collection objects of the dataset are all vehicles, pedestrians, etc., in urban scenes, their movement speeds relative to the ground should follow a similar physical distribution (such as vehicle speed limit, pedestrian walking speed). The author needs to prove that the "absolute motion speed" (rather than pixel displacement) of the target in the MMOT is indeed significantly higher than that in other datasets; otherwise, "3.4×/11.9× higher movement "may only reflect the movement of the unmanned aerial vehicle itself (such as violent rotation/translation) or annotation differences.
Potential impact of Annotation Methods (Rotating Box vs General Rectangular Box) : The low IoU may stem from the sensitivity of the rotating frame to the rapidly rotating target (the sudden drop in the overlap rate due to the change in the frame direction), while the common rectangular frame is insensitive to rotation. A comparative experiment is required: After converting the rotating box of MMOT to a general rectangular box and recalculating the IoU, observe whether it is still significantly lower than that of other datasets. If the difference disappears, it indicates that the current low IoU is artificially caused by the annotation method rather than the real motion characteristics.
Improvement suggestions: Supplementary Analysis 1: Quantify the contribution ratios of the self-movement of the unmanned aerial vehicle and the movement of the target relative to its size, and clarify the main reasons for the low IoU. Supplementary Analysis 2: Provide a comparison experiment of IoU between the rotating box and the general rectangular box to eliminate the deviation caused by the annotation form.

If the authors can answer and solve these questions, I am willing to offer suggestions for the acceptance of this paper.

**Strengths Contributions:**

This paper presents a large-scale and challenging multispectral UAV MOT dataset and proposes a drone-based multi-object tracking system based on it, which is of significant importance for advancing multi-object tracking capabilities using unmanned aerial vehicles.  This dataset is collected using a drone-mounted multispectral camera with a downward-facing view, capturing real-world urban scenes across varying dates, flight altitudes, and weather conditions. It comprises 125 video sequences totaling 13.8K frames, captured at a spatial resolution of 1200 × 900 with 8 spectral bands, contains precise oriented bounding box annotation, and includes challenges such as extremely small targets, densely packed instances, severe occlusions, fast object motion, and irregular UAV motion.

---

> ### Author Rebuttal · Authors · 2025-07-31
>
> # Response to reviewer QXcp's comments
>
> Thank you for recognizing our work. We carefully address your concerns and remain open to further discussion should you have any follow-up questions.
>
>
>
> > **Q1: Although Figure 1 b shows that the spectrum can separate the target and the background, the discrimination between the targets is still very low. Can it meet the recognition requirements of low-resolution, high-density targets and complex backgrounds for long-distance targets.**
>
> Thank you for your suggestion. Multispectral imagery not only captures spatial features but also provides **pixel-wise spectral measurements**. These spectral cues complement spatial features and enhance inter-target discriminability, where appearance cues alone are insufficient. To verify this hypothesis, we conducted the following experiments to quantitatively assess the impact of spectral information on target discriminability in UAV-based multi-object tracking.
>
> **1. Association Metrics Improvement Using Multispectral Inputs**
>
> Target discriminability ensures accurate target identification across frames, thereby enhancing association performance. Therefore, we evaluated models under both RGB and MSI inputs, using HOTA and its two sub-metrics: DetA and AssA. DetA measures detection accuracy, while **AssA evaluates how accurately predicted trajectories align with ground-truth trajectories and how consistently they maintain target identities across frames**.
>
> As shown below, both models exhibit consistent improvements when using multispectral inputs. Notably, besides HOTA and DetA, **the AssA metric improves by approximately 3.0 points**. These results suggest that spectral information particularly benefits increasing discriminability between different targets.
>
> | Model  | Input | HOTA(↑) | DetA(↑) |    AssA(↑)     |
> | :----: | :---: | :-----: | :-----: | :------------: |
> | MOTRv2 |  RGB  |  50.5   |  39.6   |    **65.6**    |
> |        |  MSI  |  54.5   |  44.1   | **68.8(↑3.2)** |
> | MeMOTR |  RGB  |  47.0   |  32.9   |    **67.9**    |
> |        |  MSI  |  50.9   |  37.1   | **70.9(↑3.0)** |
>
> **2. Inter-ID Discriminability: Local Pairwise Cosine Distance**
>
> In multi-object tracking (MOT), the **main challenge in the association stage lies in the similarity of same-class targets within local regions**. With the introduction of spectral information, the discriminability among these same-class targets can be significantly enhanced.
>
> To validate this opinion, in each frame, we computed the cosine distance between all pairs of targets belonging to the same class, considering only those pairs whose spatial distance was smaller than a predefined threshold. Finally, we averaged the cosine distances over all such pairs to obtain the **AvgCosDist** metric. This metric reflects the **degree of separability**: a higher value indicates stronger spectral distinctiveness between targets, whereas a lower value  implies that different targets are more likely to be confused.
>
> Specifically, we report results for a threshold of 200px. As shown in the table, MSI improves the average cosine distance by **150% compared to RGB**, demonstrating that spectral information substantially increases inter-target separability.
>
> |    Metric     | $d_{th}$ |  RGB  |     MSI      |
> | :-----------: | :------: | :---: | :----------: |
> | AvgCosDist(↑) |  200px   | 0.016 | 0.040(↑150%) |
>
> **Conclusion:**
>
> The above experiments—from model-level tracking metrics and inter-ID spectral discriminability—jointly demonstrate that the integration of spectral features improves **target-to-target separability**. Furthermore, as you pointed out, **Figure 1(b)** clearly shows that spectral information also enhances **target–background discrimination**. These properties make multispectral information an effective complement to spatial features, thereby offering a robust foundation for addressing real-world challenges.
>
>
>
> > **Q2: The author needs to prove that the "absolute motion speed" (rather than pixel displacement) of the target in the MMOT is indeed significantly higher than that in other datasets.**
> >
> > **Supplementary Analysis 1: Quantify the contribution ratios of the self-movement of the unmanned aerial vehicle and the movement of the target relative to its size, and clarify the main reasons for the low IoU.**
> >
> > **Supplementary Analysis 2: Provide a comparison experiment of IoU between the rotating box and the general rectangular box to eliminate the deviation caused by the annotation form**.
>
> Thank you for your insightful suggestions. We conducted additional analyses to clarify the causes behind the observed low inter-frame IoU values.
>
>
>
> **(Q2.1) Prove that the "absolute motion speed" of the target in the MMOT is indeed higher that others.**
>
> We sincerely apologize for the confusion caused by our previous use of the term *"inter-frame movement"*. What we actually intended to convey was *"inter-frame displacement"*.  And we fully agree with your suggestion to decompose this motion into UAV-induced motion and target self-motion for a more concrete analysis.
>
> Following your suggestion, we decomposed the motion components using KLT optical flow to separately model **camera-induced motion** and **object self-motion**. Based on this decomposition, we computed the **average target displacement**, **average relative displacement**, and **average inter-frame IoU** under each motion component.
>
> As shown in the table below, compared to UAVDT-MOT and VisDrone-MOT, MMOT exhibits  **higher displacement from object-self motion** (4.37 vs. 1.17 and 2.85) , **higher relative displacement (0.22 vs. 0.04 and 0.05) ** and **lower inter-frame IoU** (0.68 vs. 0.91 and 0.85). Furthermore, MMOT also demonstrates substantially **larger displacement from UAV motion**(14.12 vs. 1.46 and 2.31). These results confirm that MMOT is more challenging than existing UAV datasets, requiring advanced motion compensation and association strategies.
>
> |            Metric             |  Motion Component   | UAVDT-MOT | VisDrone-MOT | MMOT(Ours) |
> | :---------------------------: | :-----------------: | :-------: | :----------: | :--------: |
> |     Avg. Displacement(↑)      | UAV Platform Motion |   1.46    |     2.31     | **14.12**  |
> |                               | Object Self-Motion  |   1.17    |     2.85     |  **4.37**  |
> |                               |    Total Motion     |   1.29    |     4.23     | **14.43**  |
> | Avg. Relative Displacement(↑) | Object Self-Motion  |   0.04    |     0.05     |  **0.22**  |
> |                               |    Total Motion     |   0.04    |     0.08     |  **0.82**  |
> |    Avg. Inter-frame IoU(↓)    | Object Self-Motion  |   0.91    |     0.85     |  **0.68**  |
> |                               |    Total Motion     |   0.91    |     0.88     |  **0.30**  |
>
>
>
> **(Q2.2) Clarify the main reasons for the low IoU.**
>
> To gain deeper insights into the low IoU distribution, we further analyzed inter-frame IoU  by object categories, as detailed below.
>
> Due to UAV platform motion, all objects experience a global displacement component, which has a greater impact on smaller targets. And pedestrians, with an average size of only 10.8 pixels, exhibit a substantial increase in relative displacement (from 0.31 to 1.30). This results in a dramatic drop in IoU for pedestrians (from 0.54 to 0.11 for total motion). Since pedestrians are prevalent in urban scenes, their degraded IoU strongly influences the overall IoU distribution, ultimately leading to a large proportion of extremely low IoU values (below 0.1) in the dataset.
>
> |        Metric         |  Motion Component  | Ped. | Car  | Bike | Awn. | Van  | Trk. | Tri. | Bus  |
> | :-------------------: | :----------------: | :--: | :--: | :--: | :--: | :--: | :--: | :--: | :--: |
> |     Size (pixels)     |         -          | 10.8 | 46.5 | 17.8 | 21.7 | 44.7 | 58.1 | 29.6 | 73.4 |
> | Relative Displacement |    Total Motion    | 1.30 | 0.33 | 0.89 | 0.66 | 0.33 | 0.34 | 0.42 | 0.23 |
> |                       | Object Self-Motion | 0.31 | 0.11 | 0.27 | 0.19 | 0.11 | 0.18 | 0.08 | 0.12 |
> |    Inter-frame IoU    |    Total Motion    | 0.11 | 0.51 | 0.21 | 0.27 | 0.51 | 0.52 | 0.41 | 0.61 |
> |                       | Object Self-Motion | 0.54 | 0.84 | 0.65 | 0.71 | 0.83 | 0.79 | 0.83 | 0.83 |
>
>
>
> **(Q2.3) Comparison experiment of IoU between the oriented box and rectangular box**
>
> To verify whether the use of oriented bounding boxes (OBB) is responsible for the observed low IoU compared to axis-aligned boxes, we conducted a comparative experiment. Specifically, we analyzed the distribution of average inter-frame IoU under different motion components for oriented bbox annotation and axis-aligned bbox annotation. As shown in table below. The results indicate that while axis-aligned boxes yield slightly higher IoUs (a margin of only 0.01–0.04), the distributions across different motion attributes remain highly consistent with those observed when using OBB. **These findings confirm that the low IoU is not caused by the OBB annotation scheme.**
>
> |  Annotation Type  | Inter-frame IoU (Object Self-Motion) | Inter-frame IoU (Total Motion) |
> | :---------------: | :----------------------------------: | :----------------------------: |
> |   Oriented BBox   |                 0.68                 |              0.30              |
> | Axis-Aligned BBox |             0.69(↑0.01)              |          0.34(↑0.04)           |
>
>
>
> > **Dataset Code Comments: The ReadMe needs to supplement the description of the dataset in the dataset URL.**
>
> Thank you for your suggestion. Due to the constraints of the rebuttal phase, we are currently unable to update the dataset repository. However, we promise to revise and enrich the README file once the review process concludes.
>
>
>
> We sincerely appreciate your constructive feedback again and hope our response has addressed your concerns.

---

> > ### Comment · Area_Chair_4h4T · 2025-08-05
> >
> > Dear Reviewers
> >
> > Thanks for contributing to Neurips. We have received new information from the PCs:
> > “Reviewers must participate in discussions with authors before submitting “Mandatory Acknowledgement”. ” “To facilitate discussions, we extend Author-Reviewer discussions by 48h till Aug 8, 11.59pm AoE. ”
> >
> > As informed, please engage in discussions with the authors.
> >
> > Kind regards
> > AC

---

> > ### Comment · Reviewer_QXcp · 2025-08-05
> >
> > Thank you for the response. Most of my concerns were addressed

---

> > > ### Author Response · Authors · 2025-08-06
> > >
> > > We sincerely appreciate your great efforts in reviewing this paper and your recognition of our work.

---

### Official Review · Reviewer_vhgc · 2025-07-11

**Rating:** 4
**Confidence:** 3

**Summary:**

This paper introduces MMOT, the first benchmark dataset for multi-object tracking (MOT) using drone-based multispectral imagery, containing 125 sequences (13.8K frames, 488.8K annotations) with oriented bounding box (OBB) annotations for 8 object categories. The authors also propose a multispectral adaptation framework featuring: 1) a lightweight Spectral 3D-Stem module for spectral-spatial feature extraction, 2) orientation-aware Kalman filtering, and 3) an end-to-end orientation-sensitive Transformer architecture. Experiments demonstrate that multispectral input significantly boosts tracking performance, especially for small objects in dense scenes, achieving up to a 7.3% HOTA improvement.

**Dataset Code Accessibility:**

Yes

**Ethical Considerations:**

No, there are no or only very minor ethics concerns

**Limitations Weaknesses:**

1. The evaluation lacks comparisons against specialized multispectral trackers and cross-dataset testing (e.g., on MUST), limiting benchmark relevance.
2. Manual OBB labeling is labor-intensive, necessitating exploration of semi-automatic methods like active learning or weak supervision.

**Strengths Contributions:**

1. This paper establishes the first MOT dataset combining multispectral imagery and OBB annotations for drones, overcoming limitations of existing datasets that only support RGB and axis-aligned boxes.
2. This dataset features high-quality, challenging data with precise annotations developed through 5,000+ human hours and a 4-stage QA pipeline, ensuring spatiotemporal consistency. It covers diverse urban/rural/transit scenarios, including small objects, high density, occlusion, and complex motion.
3. This dataset uses Conv3D + Depthwise Conv3D to extract spectral features while maintaining compatibility with RGB pretraining, boosting convergence and performance.

---

> ### Author Rebuttal · Authors · 2025-07-31
>
> # Response to reviewer vhgc's comments
>
> Thank you for recognizing our work. We carefully address your concerns below and clarify our motivations.
>
> > **Q1: The evaluation lacks comparisons against specialized multispectral trackers and cross-dataset testing (e.g., on MUST).**
>
> Thank you for your suggestion. In our work, we propose MMOT—the first benchmark dataset designed specifically for multispectral MOT from aerial perspectives, with oriented bounding box (OBB) annotations. Our design is motivated by two key considerations:
>
> - **Multispectral input**: Spectral inputs provide complementary cues beyond spatial features, improving object discrimination and association especially under challenging scenario such as small objects and cluttered backgrounds.
> - **Oriented bounding boxes**: OBBs can represent arbitrary object orientations in aerial views, enabling more precise target representation and reducing both inter-object and inter-frame ambiguity, as also noted by reviewer QRJ8.
>
> These considerations underscore the necessity and relevance of our dataset design. To the best of our knowledge, this is the first trial to build a benchmark that jointly supports multispectral inputs and OBB-based MOT.
>
> **(Q1.1) On Comparison with Specialized Multispectral Trackers**
>
> Following your suggestion, we surveyed existing specialized multispectral trackers. And to the best of our knowledge, there are currently **no specialized multispectral trackers for MOT**, making it infeasible to compare our method against existing specialized multispectral MOT approaches.
>
> And as shown in the table below, all of the specialized multispectral trackers are developed for **Single Object Tracking (SOT)** rather than MOT. SOT algorithms rely on a target template to initialize the tracker and then estimate the location of this single target frame by frame. In contrast, our dataset is specifically designed for the **Multi-Object Tracking (MOT)** task, which involves **joint detection and tracking of multiple objects**, requiring the model to automatically detect and associate all targets in each frame without any predefined template. Therefore SOT is **fundamentally incompatible with the MOT settings**. Moreover, these SOT methods are designed for axis-aligned bounding boxes, whereas our dataset requires trackers to output **oriented bounding boxes**. These mismatches prevent the evaluation of these specialized multispectral SOT trackers in the MOT setting.
>
> We believe that comparisons against specialized multispectral trackers is important, but the limitations mentioned above collectively make it infeasible.
>
>
> |   Method   |   Year    | Spectral Input | MOT  | Oriented BBox |
> | :--------: | :-------: | :------------: | :--: | :-----------: |
> | CCTrack[1] | TGRS 2025 |       ✔️        |  ❌   |       ❌       |
> |  HDSP[2]   | TGRS 2025 |       ✔️        |  ❌   |       ❌       |
> |  SSTCF[3]  | TGRS 2025 |       ✔️        |  ❌   |       ❌       |
> | UNTrack[4] | CVPR 2025 |       ✔️        |  ❌   |       ❌       |
>
> **(Q1.2) On Cross-dataset Testing**
>
> We appreciate your suggestion. As noted in Q1.1, there are currently no existing methods specifically designed for multispectral MOT. Therefore, we adapted existing RGB-based MOT methods using our proposed **multispectral and orientation-aware MOT scheme**. These adapted models can accept **8-channel multispectral inputs** and output **oriented bounding boxes**, enabling training and evaluation on our dataset.
>
> Existing MOT datasets such as UAVDT and VisDrone only provide **RGB inputs** and **axis-aligned bounding boxes**. Because the models trained on our dataset is built to process 8-channel multispectral data, they **cannot accept 3-channel RGB inputs**. Furthermore, the difference between **oriented and horizontal bounding boxes** prevents our model from producing compatible outputs.
>
> On the other hand, as noted in Q1.1, although some SOT datasets (e.g., MUST, HOTC) provide multispectral inputs, their single-object template-based tracking paradigm is **fundamentally incompatible with the MOT setting**. This difference in task definition, combined with the mismatch between oriented and horizontal bounding boxes, makes cross-dataset evaluation infeasible.
>
> Therefore, while we fully acknowledge the importance of cross-dataset testing, the lack of multispectral MOT datasets and the paradigm gap between SOT and MOT currently make such experiments infeasible.
>
> |   Dataset    | Task | Spectral Data | Oriented Bbox |
> | :----------: | :--: | :-----------: | :-----------: |
> |  UAVDT-MOT   | MOT  |       ❌       |       ❌       |
> | VisDrone-MOT | MOT  |       ❌       |       ❌       |
> |    MOT20     | MOT  |       ❌       |       ❌       |
> |  DanceTrack  | MOT  |       ❌       |       ❌       |
> |  SportsMOT   | MOT  |       ❌       |       ❌       |
> |     HOTC     | SOT  |       ✔️       |       ❌       |
> |     MUST     | SOT  |       ✔️       |       ❌       |
> |  MMOT(Ours)  | MOT  |       ✔️       |       ✔️       |
>
> > **Q2: Manual OBB labeling is labor-intensive, necessitating exploration of semi-automatic methods like active learning or weak supervision**
>
> We fully agree with your point. Manually annotating such a large-scale dataset is indeed extremely labor-intensive, requiring more than 5,000 human-hours in our case. Our primary motivation for this effort is to build a **precisely annotated dataset**. And widely recognized as essential, precise annotations play a critical role in developing models and conducting fair benchmarking.
>
> At the same time, to minimize the human cost while maintaining high annotation quality, we made two key efforts: developing a specialized annotation tool and leveraging model-assisted labeling. We also plan to open-source this customized annotation tool to benefit the research community.
>
> - **Customized annotation tools**. Building upon X-Anylabeling[5], we developed a specialized toolset tailored for oriented bounding box multi-object tracking annotation. Compared with the original tool, our customized toolset provides the following advantages:
>
> |          Feature          | X-Anylabeling[5] | Our Tool |
> | :-----------------------: | :--------------: | :------: |
> |      Box Annotation       |        ✔️         |    ✔️     |
> |     Identity Labeling     |        ✔️         |    ✔️     |
> |  Duplicate ID Detection   |        ❌         |    ✔️     |
> | Category Change Detection |        ❌         |    ✔️     |
> |  Disappearance Detection  |        ❌         |    ✔️     |
> |  Reappearance Detection   |        ❌         |    ✔️     |
> |     ID Batch Editing      |        ❌         |    ✔️     |
>
> - **Model-assisted five-stage annotation pipeline**. As outlined in Appendix A.2, our pipeline integrates model-assisted steps for box initialization and identity suggestions to reduce human workload while ensuring expert-level accuracy. Specifically, we leverage a detection model for coarse bounding box proposals and an association model for preliminary identity assignments. These outputs are then manually aligned and refined by annotators,  which substantially reduces the annotation burden in the two most labor-intensive stages while preserving **high-quality labels**.
>
> Finally, as you suggested, we also recognize the potential of **semi-automatic annotation methods** such as weak supervision and active learning. We plan to actively explore these approaches in future work to further reduce annotation cost without compromising quality.
>
>
>
> We sincerely thank you again for your constructive feedback. We hope that this clarification resolves your concerns, and we remain open to any further discussion if needed.
>
>
>
> **References:**
>
> [1] Wang Y,et al. Hyperspectral Object Tracking With Context-Aware Learning and Category Consistency[J]. IEEE Transactions on Geoscience and Remote Sensing, 2025.
>
> [2] Yao R, et al. Hyperspectral object tracking with dual-stream prompt[J]. IEEE Transactions on Geoscience and Remote Sensing, 2024.
>
> [3] Xiong F, et al. Spatial-Spectral-Temporal Correlation Filter for Hyperspectral Object Tracking[J]. IEEE Transactions on Geoscience and Remote Sensing, 2025.
>
> [4] Qin H,, et al. MUST: The First Dataset and Unified Framework for Multispectral UAV Single Object Tracking[C]//Proceedings of the Computer Vision and Pattern Recognition Conference. 2025
>
> [5] Wei Wang. Advanced auto labeling solution with added features. https://github.com/VHub520/X-AnyLabeling, 2023.

---

> > ### Comment · Area_Chair_4h4T · 2025-08-04
> >
> > Dear Reviewer
> >
> > The Author-Reviewer discussion phase is until 6 Aug. The author has submitted a rebuttal. Please feel free to initialize discussions with the authors.
> >
> > Kind regards
> > AC

---

### Note · Authors · 2025-08-14

We sincerely thank the Program Chairs, Senior Area Chairs, Area Chairs, and all reviewers for their constructive feedback and active engagement during the rebuttal and discussion phases.

This work presents MMOT, the **first large-scale and challenging benchmark for drone-based multispectral multi-object tracking**. In contrast to existing RGB-based MOT datasets, the additional spectral information provided by MMOT helps to more effectively address real-world challenges such as small targets, complex backgrounds, and severe occlusions. The dataset also includes precise oriented bounding box annotations that are particularly well-suited for MOT tasks under aerial views. We believe that MMOT can provide valuable data support for advancing research in this field.

During the rebuttal phase, we thoroughly addressed the reviewers’ concerns, and **all reviewers acknowledged the contribution of this work and provided consistent positive assessments of its value**.

We expect that MMOT will promote further research in the field of drone-based MOT.

---

### Decision · Program_Chairs · 2025-09-18

**Decision:**

Accept (poster)

**Comment:**

This paper introduced a multispectral image dataset, which is acquired from UAV(drones),and aims for applications in multi-object tracking. All the four reviewers find the dataset novel, meaningful and technical sound and recommend acceptance. There were several minor comments, which are addressed through the rebuttals, reviewer-author discussions. In conclusion, this paper is suitable for acceptance.